climatology/statistics/biomedical engineering

regression analysis, autoregressive processes, prediction methods, supervised learning, trend estimation, statistical significance

**Author for correspondence:**
S. J. Salamon
e-mail: stephen.salamon@adelaide.edu.au

# How real are observed trends in small correlated datasets?

## S. J. Salamon, H. J. Hansen and D. Abbott

School of Electrical and Electronic Engineering, The University of Adelaide, South Australia 5005, Australia

SJS, 0000-0002-4930-3614; DA, 0000-0002-0945-2674

The eye may perceive a significant trend in plotted time-series data, but if the model errors of nearby data points are correlated, the trend may be an illusion. We examine generalized least-squares (GLS) estimation, finding that error correlation may be underestimated in highly correlated small datasets by conventional techniques. This risks indicating a significant trend when there is none. A new correlation estimate based on the Durbin–Watson statistic is developed, leading to an improved estimate of autoregression with highly correlated data, thus reducing this risk. These techniques are generalized to randomly located data points in space, through the new concept of the nearest new neighbour path. We describe tests on the validity of the GLS schemes, allowing verification of the models employed. Examples illustrating our method include a 40-year record of atmospheric carbon dioxide, and Antarctic ice core data. While more conservative than existing techniques, our new GLS estimate finds a statistically significant increase in background carbon dioxide concentration, with an accelerating trend. We conclude with an example of a worldwide empirical climate model for radio propagation studies, to illustrate dealing with spatial correlation in unevenly distributed data points over the surface of the Earth. The method is generally applicable, not only to climate-related data, but to many other kinds of problems (e.g. biological, medical and geological data), where there are unequally (or randomly) spaced observations in temporally or spatially distributed datasets.

## 1. Introduction

Ordinary least-squares (OLS) estimation, applied to time-series data, identifies a model having the lowest squared difference from the observed data. A *t*-test may be used [1] to determine the confidence interval of the regression slope, or to test its significance, but the *t*-statistic may be invalid if the residuals are correlated with each other.

This has been understood for some time [2–4], but we review existing techniques, concentrating on the generalized least-squares (GLS) formulation, and provide new techniques to make valid tests of trend or empirical model significance available for the challenging case of small datasets with strong positive error correlation. This approach is then generalized to randomly located data points in space.

We begin with a brief description of OLS and GLS, followed by an example to demonstrate problems with these conventional approaches. For the purposes of this study, we assume we are taking a limited set of samples from a field of values that have homoskedastic normally distributed error with respect to some model, with the error correlated as a function of distance in time or space between the samples.

In §2, we describe the conventional background of OLS estimation and GLS estimation of correlated time-series data, where a first-order autoregressive process is assumed to describe the residual correlation. As an example, we take a 40-year time series of atmospheric background carbon dioxide, comparing linear and parabolic regression models from conventional GLS estimation. This section concludes with some early published alternatives to the matrix algebra GLS approach.

Section 3 develops a new GLS estimate for equally spaced time-series data with highly correlated residuals. The performance of this new estimate is considered both from the point of view of providing an accurate *t*-distribution for regression slopes in the absence of any real trend, and for its ability to identify a real trend when one is present. The effect of this new estimate on the model confidence interval for our carbon dioxide example is presented.

The new GLS estimate of §3, for equally spaced data points in one-dimensional space, is generalized to randomly spaced data points in multi-dimensional space in §4, by replacing a first-order autoregressive model of error correlation with its equivalent exponential model in multi-dimensional space. We describe techniques to deal with data points co-located in that space. As an initial example, we apply this to the problem of combining two separate time-series datasets of monthly global temperature.

Section 5 provides examples applying the techniques of previous sections to real data, with two examples of time-series climatic data, both equally and unequally spaced, and an example of radio-climate data points unevenly distributed in two dimensions over the surface of the Earth.

# 2. Background

## 2.1. Ordinary least-squares estimation

Formulating OLS estimation in matrix algebra, define $\mathbf{y}$ as the $N \times 1$ vector of $N$ response or time-series values, and $\mathbf{X}$ as the $N \times k + 1$ matrix of model parameters, each row beginning with 1 (to identify the intercept) followed by $x_{1i}, \ldots, x_{ki}$. If looking for a trend in a time series, we may have $k = 1$, and only one parameter $x_{1i}$, representing time. In general, $k$ is the number of parameters in the model, not including the intercept.

If $\boldsymbol{\epsilon}$ is the $N \times 1$ vector of errors, independently normally distributed and homoskedastic, then the OLS estimator $\mathbf{b}_{\mathrm{OLS}}$ for $\boldsymbol{\beta}$ in model $\mathbf{y} = \mathbf{X}\boldsymbol{\beta} + \boldsymbol{\epsilon}$ is

$$\mathbf{b}_{\mathrm{OLS}} = (\mathbf{X}'\mathbf{X})^{-1}\mathbf{X}'\mathbf{y}$$

with residuals

$$\mathbf{e} = \mathbf{y} - \mathbf{X}\mathbf{b}_{\mathrm{OLS}}.$$

(2.1)

## 2.2. Data with correlated error

If the residuals are correlated, this suggests correlation between the errors, although the two are not the same, as $\mathbf{b}_{\mathrm{OLS}}$ is only an estimate of $\boldsymbol{\beta}$. If the errors are correlated, a better estimator may be found, if a reasonable model for this correlation is available. A first-order autoregressive process $\mathrm{AR1}(\rho)$ is often a good choice. In this paper, we restrict our consideration to that model, and an equivalent model for datasets randomly distributed in space. This model of error correlation is a short-memory model with exponential decay, which may overestimate trend significance in some cases [5], as compared with long-memory models with hyperbolic decay. However, we find that although conventional techniques described below work well in estimating negative $\rho$ in the $\mathrm{AR1}(\rho)$ for small datasets, they provide a biased underestimate of positive $\rho$-values for small sample sizes, so in this paper, we address that issue.

The parameter $\rho$ may be estimated from the lag 1 component, $r_1$, of the autocorrelation function (ACF) of the OLS residuals. In OLS estimation, residual mean will be zero, so we may write this estimate of $\rho$ as

with

$$\left.\begin{array}{l} \hat{\rho} = r_1, \\[2em] r_j = \dfrac{\displaystyle\sum_{i=1}^{N-j}(e_i \cdot e_{i+j})}{\displaystyle\sum_{i=1}^{N} e_i^2}, \end{array}\right\} \tag{2.2}$$

where $e_i$ is the residual for observation $i$, from vector $\mathbf{e}$ in (2.1).

Alternatively, $\rho$ may be estimated from the Durbin–Watson [6,7] statistic $d$ as

with

$$\left.\begin{array}{l} \hat{\rho} = 1 - \dfrac{d}{2} = 1 - \dfrac{\text{SSRFD}}{2\text{SSR}}, \\[2em] d = \dfrac{\text{SSRFD}}{\text{SSR}} = \dfrac{\displaystyle\sum_{i=2}^{N}(e_i - e_{i-1})^2}{\displaystyle\sum_{i=1}^{N} e_i^2}, \end{array}\right\} \tag{2.3}$$

denoting SSRFD as the sum of squares of residual forward differences, and SSR as the sum of squares of residuals. A matrix formulation is available for these, by constructing the matrix $\mathbf{A}$ as

$$\mathbf{A} = \begin{pmatrix} 1 & -1 & 0 & \dots & 0 & 0 \\ -1 & 2 & -1 & \ddots & 0 & 0 \\ 0 & -1 & 2 & \ddots & 0 & 0 \\ \vdots & \ddots & \ddots & \ddots & \ddots & \vdots \\ 0 & 0 & 0 & \ddots & 2 & -1 \\ 0 & 0 & 0 & \dots & -1 & 1 \end{pmatrix}. \tag{2.4}$$

Using $\mathbf{A}$, SSRFD may be evaluated as $\mathbf{e}'\mathbf{A}\mathbf{e}$, and with SSR $= \mathbf{e}'\mathbf{e}$ we have

$$d = \frac{\mathbf{e}'\mathbf{A}\mathbf{e}}{\mathbf{e}'\mathbf{e}}. \tag{2.5}$$

## 2.3. Correlation testing the ordinary least-squares scheme

Departure of either the ACF lag terms, or the Durbin–Watson statistic $d$, of the OLS residuals from the range of variation expected for uncorrelated errors, may be used as a test for serial correlation affecting the validity of the OLS estimate. Our main concern is short-memory correlation, so we use $d$, with a confidence interval based on exact expressions for its expected value $E(d)$, and variance $V(d)$, provided by Durbin & Watson [6] for the case of uncorrelated errors:

with

and

with

$$\left.\begin{array}{l} E(d) = \dfrac{p}{N - k - 1} \\[1.5em] p = \text{tr}(\mathbf{A}) - \text{tr}[\mathbf{X}'\mathbf{A}\mathbf{X}(\mathbf{X}'\mathbf{X})^{-1}] \\[1.5em] V(d) = \dfrac{2(q - pE(d))}{(N - k - 1)(N - k + 1)} \\[1.5em] q = \text{tr}(\mathbf{A}^2) - 2\text{tr}[\mathbf{X}'\mathbf{A}^2\mathbf{X}(\mathbf{X}'\mathbf{X})^{-1}] + \text{tr}[[\mathbf{X}'\mathbf{A}\mathbf{X}(\mathbf{X}'\mathbf{X})^{-1}]^2]. \end{array}\right\} \tag{2.6}$$

Thus the distribution of $d$ depends to some extent on the data in $\mathbf{X}$. A number of methods have been proposed [7] to calculate tail-points of the distribution of $d$, but the beta approximation has been

found [7] to perform well, and is now very convenient, given the availability of functions for the inverse cumulative beta distribution in current computer languages. In this approximation, $d/4$ is assumed to have a beta distribution $\beta_{a,b}$, with mean and variance equal to $E(d)$ and $V(d)$, respectively, by setting

$$a + b = \frac{E(d)[4 - E(d)]}{V(d)} \quad \text{and} \quad a = \frac{(a + b)E(d)}{4}. \tag{2.7}$$

Taking upper and lower cumulative probabilities as 0.975 and 0.025, respectively, we find the limits of the 95% confidence interval for $d$, given the null hypothesis of no serial correlation. If $d$ falls outside this range, we conclude that either OLS estimation is unsatisfactory, and an alternative such as GLS estimation must be used, or the model being fitted to the data is unsuitable.

## 2.4. GLS estimation

If a correlation test, such as the Durbin–Watson test described above, suggests that OLS estimation is unsatisfactory, then the OLS estimator of (2.1) is modified, by pre-multiplying both $\mathbf{y}$ and $\mathbf{X}$ by a matrix $\mathbf{P}$. This matrix $\mathbf{P}$ aims to transform the error vector $\boldsymbol{\epsilon}$, with correlated elements, into vector $\mathbf{P}\boldsymbol{\epsilon}$, with uncorrelated elements of equal variance. Pre-multiplying the problem by $\mathbf{P}$ results, observing the correct order of multiplication of transposed matrices, in the GLS estimate

where

$$\left. \begin{array}{c} \mathbf{b}_{\text{GLS}} = (\mathbf{X}'\mathbf{S}^{-1}\mathbf{X})^{-1}\mathbf{X}'\mathbf{S}^{-1}\mathbf{y} \\ \\ \mathbf{S}^{-1} = \mathbf{P}'\mathbf{P}. \end{array} \right\} \tag{2.8}$$

For the AR1($\rho$) first-order autoregressive error process, with our homoskedastic assumption, we may take symmetrical matrix $\mathbf{S}$ to be the matrix of correlations between samples, as

$$\mathbf{S} = \begin{pmatrix} 1 & \rho & \rho^2 & \cdots & \rho^N \\ \rho & 1 & \rho & \cdots & \rho^{N-1} \\ \rho^2 & \rho & 1 & \cdots & \rho^{N-2} \\ \vdots & \vdots & \vdots & \ddots & \vdots \\ \rho^N & \rho^{N-1} & \rho^{N-2} & \cdots & 1 \end{pmatrix}, \tag{2.9}$$

which has a readily calculated inverse $\mathbf{S}^{-1}$, consisting of only diagonal, superdiagonal and subdiagonal elements

$$\mathbf{S}^{-1} = \frac{1}{1 - \rho^2} \begin{pmatrix} 1 & -\rho & 0 & \cdots & 0 & 0 \\ -\rho & 1 + \rho^2 & -\rho & \cdots & 0 & 0 \\ 0 & -\rho & 1 + \rho^2 & \cdots & 0 & 0 \\ \vdots & \vdots & \vdots & \ddots & \vdots & \vdots \\ 0 & 0 & 0 & \cdots & 1 + \rho^2 & -\rho \\ 0 & 0 & 0 & \cdots & -\rho & 1 \end{pmatrix}. \tag{2.10}$$

## 2.5. Maximum-likelihood estimation

A standard method of estimating $\rho$ is maximum-likelihood (ML) estimation. Here, we jointly find the most likely estimates for $\rho$ and the model parameters $\beta$ to explain the observations, assuming a Gaussian error process. In the case of the AR1($\rho$) process, numerical evaluation may be avoided by obtaining the ML estimate of $\rho$ conditioned on $e_1$ [8]:

$$\hat{\rho} = \frac{\sum_{i=2}^{N} (e_i - \bar{e}_{(2)})(e_{i-1} - \bar{e}_{(1)})}{\sum_{i=2}^{N} (e_{i-1} - \bar{e}_{(1)})^2} \tag{2.11}$$

with $\bar{e}_{(1)} = (N - 1)^{-1} \sum_{i=1}^{N-1} e_i$ and $\bar{e}_{(2)} = (N - 1)^{-1} \sum_{i=2}^{N} e_i$.

An iterative scheme, re-evaluating the residuals as $\mathbf{e} = \mathbf{y} - \mathbf{X}\mathbf{b}_{\text{GLS}}$, and re-evaluating $\hat{\rho}$ from (2.11), until it converges, may be used [9] to obtain the joint ML estimate of $\rho$ and $\beta$. However, this is only valid for equally spaced data in one dimension.

## 2.6. Correlation testing the GLS scheme

The Durbin–Watson test described in §2.3 above may be applied to the GLS residuals, pre-multiplied by $\mathbf{P}$, to test if correlation remains after pre-multiplying the problem by $\mathbf{P}$. This Durbin–Watson statistic is then

where

$$\left.\begin{array}{l} d = \dfrac{\mathbf{e}'\mathbf{P}'\mathbf{A}\mathbf{P}\mathbf{e}}{\mathbf{e}'\mathbf{S}^{-1}\mathbf{e}} \\[12pt] \mathbf{e} = \mathbf{y} - \mathbf{X}\mathbf{b}_{\mathrm{GLS}}. \end{array}\right\} \qquad (2.12)$$

Matrix $\mathbf{S}^{-1}$ only, not $\mathbf{P}$, appears in the GLS estimation $\mathbf{b}_{\mathrm{GLS}} = (\mathbf{X}'\mathbf{S}^{-1}\mathbf{X})^{-1}\mathbf{X}'\mathbf{S}^{-1}\mathbf{y}$, but $\mathbf{P}$ is generally not uniquely defined by $\mathbf{P}'\mathbf{P} = \mathbf{S}^{-1}$. As different real matrices $\mathbf{P}$ satisfying this give slightly different values of $d$, we need to choose it in a consistent way. For the GLS models described in this paper, we may choose $\mathbf{P}$ to be the real symmetric matrix that is the principal square root of $\mathbf{S}^{-1}$. The confidence interval for $d$ may be determined in the same way as before, except $p$ and $q$ are now evaluated as

$$\left.\begin{array}{l} p = \mathrm{tr}(\mathbf{A}) - \mathrm{tr}[\mathbf{X}'\mathbf{P}'\mathbf{A}\mathbf{P}\mathbf{X}(\mathbf{X}'\mathbf{S}^{-1}\mathbf{X})^{-1}] \\[8pt] q = \mathrm{tr}(\mathbf{A}^2) - 2\mathrm{tr}[\mathbf{X}'\mathbf{P}'\mathbf{A}^2\mathbf{P}\mathbf{X}(\mathbf{X}'\mathbf{S}^{-1}\mathbf{X})^{-1}] + \mathrm{tr}[[\mathbf{X}'\mathbf{P}'\mathbf{A}\mathbf{P}\mathbf{X}(\mathbf{X}'\mathbf{S}^{-1}\mathbf{X})^{-1}]^2]. \end{array}\right\} \qquad (2.13)$$

and

If $d$ falls outside the chosen confidence interval, it may indicate that $\mathbf{S}^{-1}$ does not adequately compensate for the correlation between errors, or it may indicate that the model being fitted to the data is not appropriate. We see an example of this in §2.8.

## 2.7. Alternatives to GLS

The GLS formulation above is by no means the only way of analysing time-series data with autoregressive errors. The simplest approach [3,10] is to calculate an effective number of samples $N_{\mathrm{eff}}$ in calculating the standard error of regression coefficients

$$N_{\mathrm{eff}} = N \frac{1 - \rho}{1 + \rho}. \qquad (2.14)$$

We find, by comparing this method with GLS estimation with known $\rho$, that this approach is only applicable to calculating the standard errors of the regression coefficients, not the residual standard error. A recent study, detailing the calculation of regression coefficient standard errors with this method, is provided by [11].

However, this method may fail with small datasets with highly correlated residuals; e.g. if $N = 38$ and $\rho = 0.9$, we have $N_{\mathrm{eff}} = 2$, leading to zero degrees of freedom for $k = 2$, and infinite standard errors. Even for less extreme cases, there is a risk of this approach over-compensating.

We propose a simple alternative, almost identical for large $N$, but avoiding this risk, by applying the scaling of (2.14) to the regression coefficient variance estimates, or the square root of this scaling to the OLS-calculated regression coefficient standard errors $s_{b\mathrm{OLS}}$, to obtain an estimate for GLS estimation with known $\rho$:

$$\hat{s}_{b\mathrm{GLS}} = s_{b\mathrm{OLS}}\sqrt{\frac{1 + \rho}{1 - \rho}}. \qquad (2.15)$$

The regression coefficient $t$-value is then obtained by dividing the coefficient by $s_{b\mathrm{GLS}}$ as usual. A weakness of the schemes of (2.14) and (2.15) are that they do not allow for testing of the correctness of the error correlation model, as described in §2.6 above.

Another simple scheme for dealing with correlated time-series data is Cochrane–Orcutt [4] estimation. This involves transforming the dataset by subtracting $\rho$ times the previous sample value from each current sample value in $\mathbf{y}$ and $\mathbf{X}$, to provide uncorrelated residuals in OLS estimation of the transformed dataset. The regression coefficients and $t$-values obtained converge to those of GLS estimation with known $\rho$, as $N$ becomes large compared to $-1/\ln(\rho)$, although the first observation is lost, leaving a dataset of size $N - 1$.

## 2.8. GLS estimation applied to atmospheric carbon dioxide background

As an example of GLS estimation, consider the 40-year record of monthly mean background atmospheric carbon dioxide concentration, measured at Cape Grim, Tasmania [12]. As there is a small annual

variation with the seasons in the data, we take the January to December mean for each complete year, from 1977 to 2016. Using a linear model, of the form $y = b_0 + b_1 \, t$, the residuals are very highly correlated, with an extremely low Durbin–Watson statistic of $d = 0.105$. This indicates OLS estimation with the linear model is not appropriate; this may be either due to error correlation, or due to the linear model being unsuitable, or both.

As described above, we are assuming a first-order autoregressive (AR) process for the errors, and estimate $\rho$, to perform GLS estimation. The methods for estimating $\rho$ described so far yield different results, but the Durbin–Watson tests on the transformed residuals, as described above in §2.6, do not yield results within the 95% confidence interval for uncorrelated residuals, for any of the $\rho$ estimates. The ACF lag 1 estimate is $\rho = 0.825$, giving $d = 0.860$, and the probability (two-tailed test assuming a beta distribution for $d$) of $d$ being that far removed from $E(d)$ is only 0.00003.

ML, iterated jointly for $\rho$ and $\beta$, estimates $\rho = 0.931$, giving $d = 1.167$, and two-tailed probability of 0.003, but still well outside the 95% confidence interval. The Durbin–Watson estimate $\rho = 1 - d/2$ is slightly higher with $\rho = 0.948$, giving $d = 1.212$, but the two-tailed probability is still only 0.005, well outside the 95% confidence interval. This may suggest that the first-order AR process is an unsuitable model for the errors, but an alternative explanation is that the linear trend model is unsuitable for this data.

The latter explanation tends to be supported when we try a quadratic model to fit the data, of the form $y = b_0 + b_1 \, t + b_2 \, t^2$, where the times $t$ have been shifted to zero mean. The OLS residuals now give $d = 0.850$, still indicating that GLS is required, but the estimates of $\rho$ are now smaller: 0.446 for ACF lag 1; 0.575 for Durbin–Watson $1 - d/2$, and 0.535 for MLE. Of these three, only the Durbin–Watson estimate gives a two-tailed test on the GLS residuals within the 95% confidence interval, with probability 0.065.

Only by assuming a cubic model, of the form $y = b_0 + b_1 \, t + b_2 \, t^2 + b_3 \, t^3$, do we get a satisfactory result for the MLE $\rho$. We now have $\rho = 0.518$, giving $d = 1.617$, and two-tailed probability of 0.078, or just inside the 95% confidence interval. Durbin–Watson estimation of $\rho$ yields a very similar result, with $\rho = 0.508$, giving $d = 1.604$, and two-tailed probability of 0.071. The ACF lag 1 estimate is now $\rho = 0.455$, yielding a still unsatisfactory $d = 1.537$, and two-tailed probability of 0.042. All these GLS estimates yield a significant $t$-value for the $b_3$ coefficient, in the region of 2.7, indicating the cubic model is more appropriate than linear or quadratic models.

The quadratic model for the Durbin–Watson estimate $\rho = 1 - d/2$ and the cubic model for the ML estimate of $\rho$ are shown in figure 1. The cubic model for the Durbin–Watson estimate is not shown, as it is almost identical to MLE. Prediction intervals, as defined in §2.9 below, are shown. However, the size of these prediction intervals may be underestimated, since the Durbin–Watson test on the transformed residuals in both cases only just falls within the 95% confidence interval.

## 2.9. The prediction interval—supervised machine learning

The prediction interval for OLS estimation, with a linear model of one parameter ($k = 1$), is defined in texts [1], so consistent with that definition, we define it here for all $k$, and GLS estimation. If the best-fit curve and prediction interval are evaluated in the region beyond the available data, we may refer to it as *supervised machine learning*. If the mean prediction at $[x_{02}, \ldots, x_{0k+1}]$ is $\hat{y}_0$, then the $100(1 - \alpha)\%$ prediction interval of $y_0$ is

$$\hat{y}_0 - t_{\alpha/2} s_p < y_0 < \hat{y}_0 + t_{\alpha/2} s_p,$$

where

$$s_p = \sqrt{s^2 + s_{b_1}^2 + \sum_{i=2}^{k+1} s_{b_i}^2 (x_{0i} - \bar{x}_i)^2},$$

and for GLS,

$$s = \sqrt{\frac{\mathbf{e}' \mathbf{S}^{-1} \mathbf{e}}{N - k - 1}}$$

and

$$s_{b_i} = s \sqrt{[(\mathbf{X}' \mathbf{S}^{-1} \mathbf{X})^{-1}]_{ii}}.$$

(2.16)

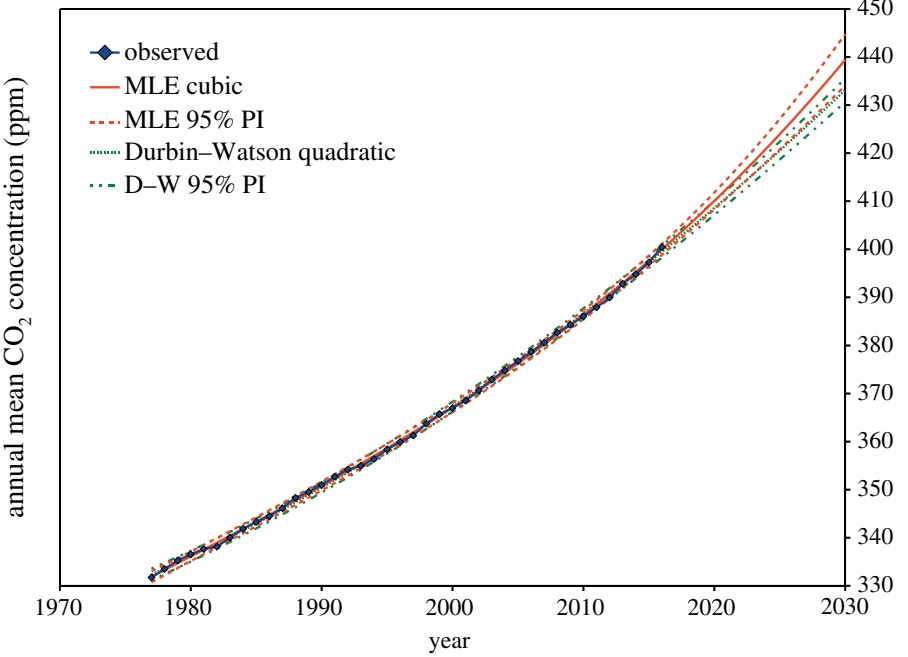

**Figure 1.** Annual means of background atmospheric carbon dioxide, Cape Grim, Tasmania. A quadratic GLS model, with $\rho = 1 - d/2$ estimated from the Durbin–Watson statistic $d$ is shown in green, and a cubic model, by iterative joint maximum-likelihood estimation of $\rho$ and $\beta$ is shown in red. For each estimation procedure, this is the minimum order polynomial passing the Durbin–Watson test on transformed residuals. The 95% prediction interval is shown for both of these models in dashed or dashed and dotted lines; these intervals and the fitted models are shown extended to the year 2030.

In (2.16), $t_{\alpha/2}$ is the upper $\alpha/2$ tail-point of the $t$-distribution with $N - k - 1$ degrees of freedom, $s$ is the residual standard error, and the $s_{b_i}$ are the standard errors of the elements of $\mathbf{b}_{GLS}$. This expression assumes the explanatory variables in $\mathbf{X}$ are uncorrelated. This may be achieved when the data is assembled in $\mathbf{X}$ by subtracting from each new variable its OLS estimate in terms of the previous variables. This decorrelation procedure is only required for an accurate estimate of the prediction interval by (2.16), but has no effect on OLS or GLS estimation $\mathbf{y}$.

However, (2.16) assumes the GLS transformation defined by $\mathbf{S}^{-1}$ accurately accounts for the error correlation. If the model of error correlation is an estimate, (2.16) may underestimate the prediction interval. Figure 2 demonstrates this potential problem, using both quadratic and cubic MLE fits to the first 25 years of the Cape Grim carbon dioxide background data, and using the subsequent 15 years to test the prediction interval. Training from just the first 25 years, a quadratic model with OLS estimation still does not pass the Durbin–Watson test on its residuals, but all GLS methods mentioned above do pass this test on their transformed residuals.

In figure 2, we see that two of the 15 test points (13%) fall outside the MLE 95% prediction interval. A cubic model of carbon dioxide background concentration is not justified by the first 25 years of data, as the $b_3$ regression coefficient is not statistically significant ($t = 0.58$), resulting in an excessive prediction interval in the forecast region. Thus with MLE, as defined by (2.11), neither of these forecasts appear completely satisfactory. Very similar results are obtained with the other GLS methods described above; ACF in (2.2), and Durbin–Watson in (2.3).

# 3. GLS analysis of highly correlated data

## 3.1. A fundamental requirement

Our example of carbon dioxide measurements demonstrates highly correlated residuals, especially in the case of an ill-fitting regression model. It has been suggested [13] that a number of climatic parameters may naturally have a spectral characteristic with a slope approximating $f^{-2}$, or in terms of a first-order autoregressive process, $\rho$ approaching +1. Depending on the type of model chosen to fit the data, we have seen that high correlation between residuals of adjacent data samples may be present. Assuming

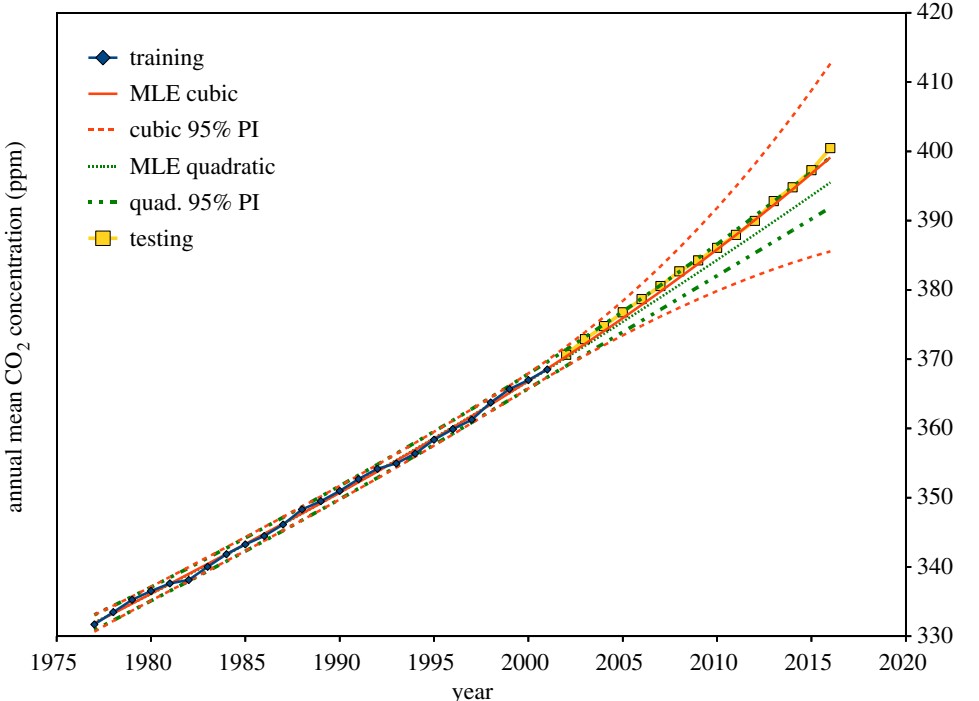

**Figure 2.** Annual means of background atmospheric carbon dioxide, Cape Grim, Tasmania. A quadratic GLS model is shown in green, and a cubic model in red, both by iterative joint maximum-likelihood estimation of $\rho$ and $\beta$. Prediction intervals (95% confidence), as defined by (2.16), are shown for both. The last two of the 15 test points fall outside the prediction interval for the quadratic model, while the prediction interval for the cubic model diverges due to the large standard error of the statistically insignificant time-cubed coefficient. Forecasting, by extending the prediction interval beyond the available data, assumes both the suitability of the fitted model and the stationarity of the process.

a first-order AR model of error correlation to be the cause, estimates of $\rho$ in (2.2), (2.3) and (2.11) may underestimate $\rho$ as it approaches +1, especially if the number of data points, $N$, is not large.

A prime requirement of the GLS estimate is that for the null hypothesis, i.e. when there is no real trend in the data, the estimated trend divided by the standard error of that trend, should follow some known distribution. If $\rho$ is known, this ratio is the $t$-distribution for $N - k - 1$ degrees of freedom, as demonstrated by the dark blue curves in figure 3. If $\rho$ is estimated rather than known, the distribution of this ratio will be a different distribution. Figure 3 demonstrates this for the non-iterated Yule–Walker (ACF lag 1) and Durbin–Watson estimates of $\rho$, as well as the iterated ML joint estimate of $\rho$ and $\beta$ of (2.11). All of these depart significantly from the $t$-distribution for $N - k - 1$ degrees of freedom, though the agreement between the non-iterated Durbin–Watson estimate and the iterated ML estimate is surprisingly close.

## 3.2. Improving the estimate of $\rho$

The curves of figure 4 show the mean value of various estimates of $\rho$, for different sample sizes $N$, with 1000 trials with AR($\rho = 0.9$) noise for each $N$. The 95% confidence interval for the ML case is indicated by the light green error bars. These curves suggest the differences between $t$-statistics with known $\rho$ and estimated $\rho$, in figure 3, may be partly due to a bias towards underestimating $\rho$ as it approaches 1.

A new estimate of $\rho$, with similar variance to MLE, but with greatly reduced bias for positive values approaching 1, is available by considering Durbin and Watson's exact expressions for $E(d)$ and $V(d)$ in the residuals of OLS estimation, for the $\rho = 0$ case, shown above at (2.6). The general idea is first to subtract this $E(d)$ from $d$, to give $d' = d - E(d)$, before calculating $\hat{\rho} = 1 - d'/2$, to ensure $\hat{\rho}$ is always unbiased for the $\rho = 0$ case. This still leaves a variance in $\hat{\rho}$ that becomes unduly compressed for small $N$, so we scale the variance to reduce the bias for non-zero $\rho$.

The remaining problem is that $\hat{\rho}$ must be restricted to the range $-1 < \hat{\rho} < 1$. As the beta distribution has been found [7] to be a good approximation to the distribution of $d$, an obvious way to restrict the

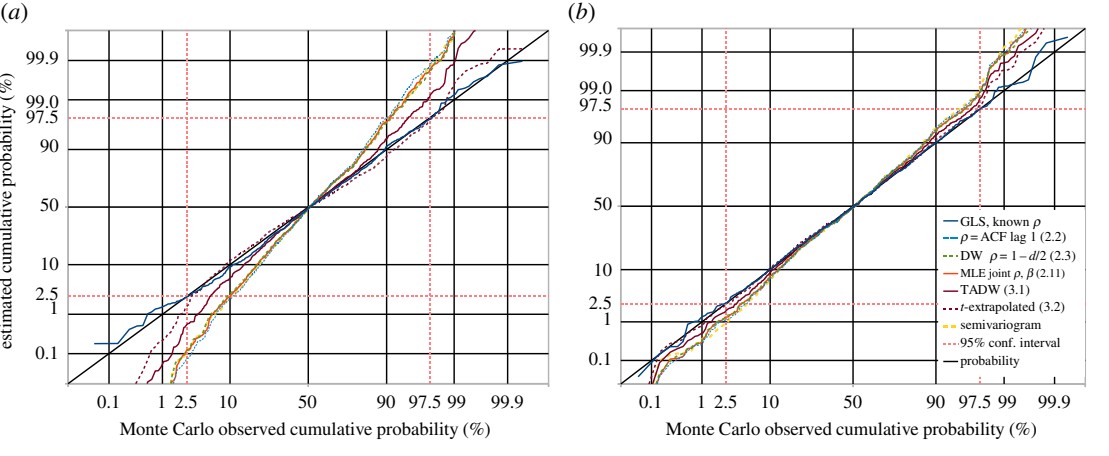

**Figure 3.** 1000 trials of GLS slope-$t$ estimation, with datasets of autoregressive random noise. The dark blue solid line is the cumulative distribution function (CDF) of $t$-values produced by GLS analysis using known $\rho$, showing excellent agreement with the $t$-distribution with $N - k - 1$ degrees of freedom (the diagonal black line). Estimated $\rho$-values give poorer agreement: the Yule–Walker estimate from the ACF of residuals (light blue dotted line); from the Durbin–Watson statistic (green dashed line); iterative joint maximum-likelihood estimation of $\rho$ and $\beta$ (the continuous red line, almost co-incident with the Durbin–Watson dashed line), and the semivariogram estimate described in §4. The continuous dark brown line is our improved Durbin–Watson estimate of $\rho$. Extrapolating from the two Durbin–Watson derived estimates is the dashed dark brown line; this provides accurate 95% confidence intervals for all $\rho < 0.8[(N/100)^{0.07}]$. (a) 40-point datasets of AR1(0.75) noise. (b) 250-point datasets of AR1(0.85) noise.

range was by means of a transform and inverse transform using the cumulative beta distribution, but this proved to be numerically impractical for $\rho$ well removed from zero. Instead, we use a hyperbolic tangent transform, to predict $\rho$ from the Durbin–Watson statistic of OLS residuals $d$, and expectation $E(d)$ and variance $V(d)$ from (2.6):

$$\hat{\rho} = \text{real}\left[\tanh\left(\left[\text{arctanh}\left(1 - \frac{d}{2}\right) - \text{arctanh}\left(1 - \frac{E(d)}{2}\right)\right]\frac{2}{N-k-4}\sqrt{\frac{N-k+2}{V(d)}}\right)\right]. \qquad (3.1)$$

We refer to this as the tanh adjusted Durbin–Watson (TADW) estimate of $\rho$. Although (3.1) is theoretically real, in practice, slight numerical error sometimes generates a miniscule imaginary component, which must be removed to eliminate problems in subsequent calculations.

As $N \to \infty$, in (2.6), $p \to 2(N-k)$ and $q \to 6N - 8k$, so if $N \gg k$, as $N \to \infty$, $E(d) \to 2$, $V(d) \to 4/N$, and in (3.1) $\hat{\rho} \to 1 - d/2$. Thus $\hat{\rho}$ from (3.1) is asymptotically accurate, as are the other estimates in figure 4.

The red curve in figure 4 demonstrates the mean $\rho$ prediction performance of TADW, together with 95% error bars, which are of similar magnitude to the MLE error bars. However, the bias in the TADW prediction is much less than MLE, so the estimate of $\rho$ from (3.1) is generally a better estimate than provided by ML.

## 3.3. Accurate significance testing

Although the TADW estimate of $\rho$ in (3.1) is less biased than the other estimates in figure 4, the calculated $t$-statistic for the regression slope still does not accurately follow the $t$-distribution for $N - k - 1$ degrees of freedom when $N$ is relatively small, as demonstrated by the continuous brown line in figure 3a. This is to be expected, but it would be convenient to have an estimated $t$-value following the $N - k - 1$ degrees of freedom distribution, for consistency with the known $\rho$ case. From testing of many different cases, we find that a $t$-value reasonably accurately achieving this in all but the extreme tails of the distribution is obtained by simple extrapolation from the $t$-values $t_{\text{dw}}$ obtained from GLS estimation with the simple Durbin–Watson $\rho$ estimate from (2.3), and $t_{\text{tadw}}$ obtained from GLS estimation with the improved Durbin–Watson $\rho$ estimate from (3.1). We then have $t_{\text{extrap}}$ as

$$t_{\text{extrap}} = 2t_{\text{tadw}} - t_{\text{dw}}. \qquad (3.2)$$

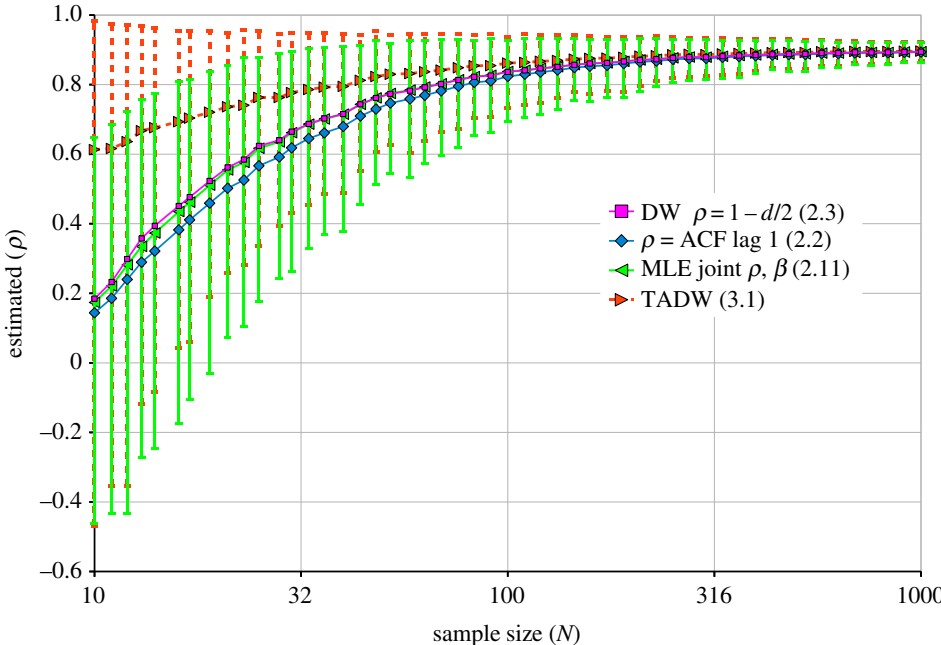

**Figure 4.** Estimated $\rho$-values for 1000 trials with AR($\rho = 0.9$) noise, at each sample size, N. The blue curve is the estimate from the OLS residual ACF; the magenta curve is from the Durbin–Watson statistic of the OLS residuals, using $\rho = 1 - d/2$. The light-green curve, with 95% confidence interval error bars, uses the iterative joint maximum-likelihood (ML) estimate of (2.7). The ML curve is surprisingly close to the Durbin–Watson curve. The red dashed curve, with dotted 95% confidence interval error bars, uses the new tanh adjusted Durbin–Watson estimate of (3.1). This figure shows the performance with one model variable, but similar results are obtained with more variables, though with slightly more bias at low N.

This $t$-value is shown as the dashed brown curves in figure 3. Our testing indicates that (3.2) reasonably accurately follows the $t$-distribution for $N - k - 1$ degrees of freedom between the 2.5% and 97.5% points of the distribution provided $\rho$ does not exceed $\rho_{\max}$, given by

$$\rho_{\max} = 0.8\left(\frac{N}{100}\right)^{0.07}. \tag{3.3}$$

Similar results, only slightly inferior, are obtained by calculating the $t$-value from the regression slope divided by an extrapolated standard error $s_{b\text{extrap}}$, similarly given by

$$s_{b\text{extrap}} = 2s_{b\text{tadw}} - s_{b\text{dw}}. \tag{3.4}$$

This extrapolated standard error $s_{b\text{extrap}}$ may then be used in constructing the prediction interval as defined in (2.16).

Separating actual trend from error correlation in small datasets is the significant challenge addressed by this study. As well as demonstrating a valid test of significance in the null hypothesis case, as shown in figure 3, we must demonstrate that real trends may be recognized with adequate sensitivity. In figure 5 we see the result of 100 tests with correlated noise plus a real trend, varying from a significant negative value, through zero, to significant positive. All methods recognize a very significant trend as such, but using MLE and assuming a $t$-distribution with $N - k - 1$ degrees of freedom overestimates significance relative to GLS with known $\rho$ for 77% of cases in this test. On the other hand, the extrapolated $t$-value from (3.2) overestimates significance relative to known $\rho$ for 38% of the time, or underestimates 62% of the time, so is slightly less sensitive to real trends than if $\rho$ is known. This is to be expected, as we are relying on an estimate of $\rho$.

## 3.4. Cape Grim carbon dioxide data; a revised trend estimate

In figure 6, we repeat the plot of figure 1, but this time, the GLS estimation uses the new TADW estimate of $\rho$ from (3.2), together with extrapolated standard errors from (3.4) to calculate the prediction intervals. As the cubic model has a statistically significant $t^3$ coefficient, we conclude, based only on

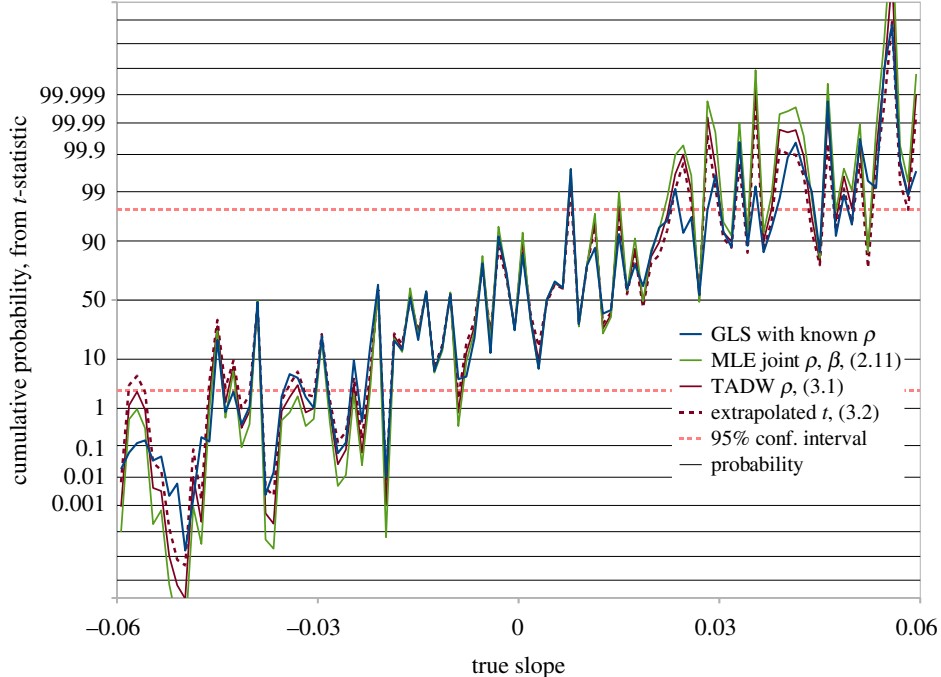

**Figure 5.** GLS regression slope $t$-values for 100 randomly generated datasets, each with 100 data points, consisting of AR1($\rho =$ 0.75) noise $\epsilon_i = N(0, 1) + \rho \cdot \epsilon_{i-1}$, plus a known slope. Here, $N(0, 1)$ is a zero mean unit variance normally distributed variate. We set $\epsilon_1 = N(0, 1)/\sqrt{1 - \rho^2}$, to ensure stationarity. The GLS calculation is performed both with known and estimated values of $\rho$. The MLE (green trace) $|t|$ exceeds the known $\rho$ $|t|$ for 77% of these trials, while the extrapolated (dashed brown trace) $|t|$ from (3.2) exceeds the known $\rho$ $|t|$ in 38% of cases.

the measured data, that the cubic model prediction interval may be more realistic than that of the quadratic model.

The actual mean prediction here for the year 2030 is 440.6 ppm, very close to the MLE cubic model prediction of 439.4 ppm, but the prediction interval is significantly wider, now 16.3 ppm between 2.5% and 97.5% points, compared with 10.7 ppm for MLE. The wider interval reflects the greater uncertainty indicated by our new model.

# 4. GLS for randomly sampled data in N-dimensional space

So far we have examined uniformly spaced data points on a one-dimensional time-line, assuming an autoregressive model for the errors, which has a particular direction in time. However the correlation matrix **S** in (2.9) is symmetric, and hence completely agnostic to the direction of time. This suggests generalizing the analysis, by considering its elements as radial basis functions of the radial distance $r_{ij}$, however defined, between the data points $i$ and $j$, of form

$$\phi_{ij} = \exp\left(-\frac{r_{ij}}{r_0}\right), \tag{4.1}$$

to give matrix **S** as

$$\mathbf{S} = \begin{pmatrix} 1 & \phi_{12} & \phi_{13} & \cdots & \phi_{1N} \\ \phi_{21} & 1 & \phi_{23} & \cdots & \phi_{2N} \\ \phi_{31} & \phi_{32} & 1 & \cdots & \phi_{3N} \\ \vdots & \vdots & \vdots & \ddots & \vdots \\ \phi_{N1} & \phi_{N2} & \phi_{N3} & \cdots & 1 \end{pmatrix}. \tag{4.2}$$

This matrix must be inverted to perform the GLS estimation of (2.8). This is generally successful, due to the exponential decay of the $\phi_{ij}$ terms with distance. If $k$ is large and correlation is considerable, an occasional problem is inversion of $(\mathbf{X}'\mathbf{S}^{-1}\mathbf{X})$, so the availability of a GLS estimate for a particular type of model is not necessarily guaranteed.

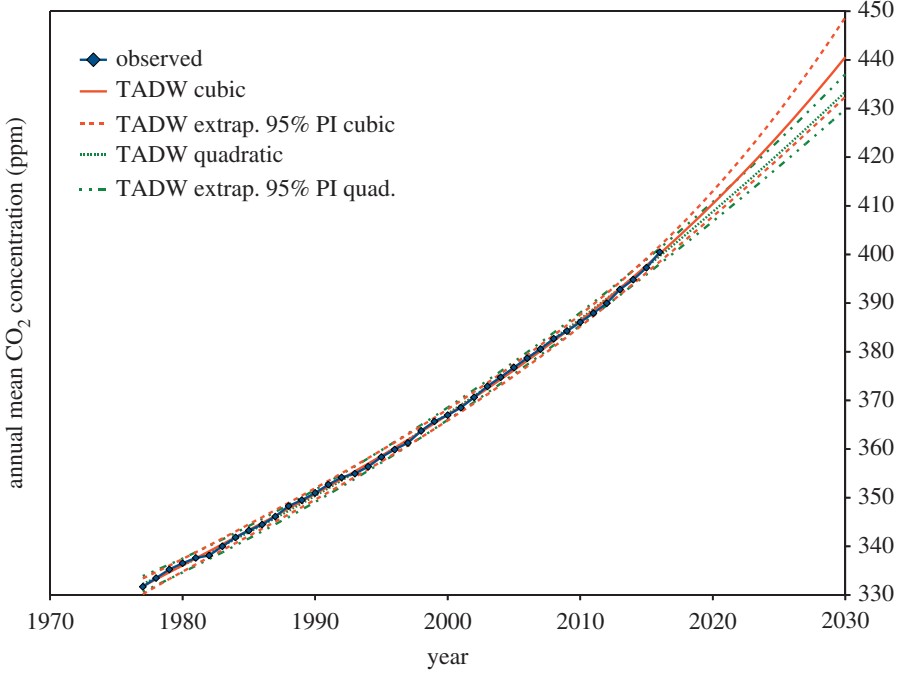

**Figure 6.** Annual means of background atmospheric carbon dioxide, Cape Grim, Tasmania. Linear and quadratic GLS models, using the new TADW estimate of $\rho$ from (3.1), are plotted. Prediction intervals are constructed with the extrapolated standard errors from (3.4). The quadratic model prediction interval from 2016 to 2030 overlaps the interval for the cubic model, but as the $t^3$ coefficient in the cubic model is just statistically significant ($t_{extrap} = 2.23$, $Pr(>|t|) = 0.032$), the wider prediction interval of the cubic model may be more realistic.

However, as we saw with the Cape Grim carbon dioxide data, residual correlation may be partly due to the choice of an unsuitable model for the data. A more suitable model may significantly reduce residual correlation, and resolve the problem if it occurs.

## 4.1. Estimating $r_0$

A useful tool to model the correlation between spatially distributed residuals is likely to be the *semivariogram*, used in *kriging* estimation [14,15]. This plots half the squared difference between all pairs of values, as a function of distance between the locations of those values, so the semivariogram plots $\gamma_{ij}$,

$$\gamma_{ij} = \frac{(e_i - e_j)^2}{2}, \qquad (4.3)$$

against distance $r_{ij}$. The aim is to find a function that is a reasonable fit to the scatter of points.

At large distances $r_{ij}$ the semivariogram is expected to tend towards a *sill* value, equal to the residual variance $\sigma_e^2$, and at smaller distances, a scatter of points around a function of the form

$$\gamma(r) = \sigma_e^2 \left[ 1 - \exp\left( -\frac{r}{r_0} \right) \right]. \qquad (4.4)$$

In kriging, this is known as the exponential model, one of several commonly used semivariogram models. We use it here partly for consistency with the first-order autoregressive model for time-series data, and partly because in kriging it tends to have a stability advantage over some other models, without resorting to the *nugget effect*, due to its large slope near the origin, so it may have a similar advantage here. In kriging, the nugget effect relaxes the constraint that the fitted surface must pass through the known points. Another reason for using the exponential model here is that it is valid in space of any number of dimensions [15].

Naturally, residual correlation in equally spaced time-series data, as discussed above, may be analysed with a semivariogram; in fact, the asymptotic expected value of the semivariogram for this

case, with first-order AR errors, is

$$E(\gamma(r)) = \sigma_e^2[1 - \rho^m] = \sigma_e^2\left[1 - \exp\left(-\frac{r}{r_0}\right)\right],$$

where $m$ is the distance in sample spacings,

and

$$r_0 = -\frac{1}{\ln(\rho)}.$$

(4.5)

Repeating the tests in figure 3, a least-squares inverse distance squared weighted fit of class means (one $m$-value per class) proved to be an effective estimate of $\rho$, quite accurate for large $N$. However, these semivariogram results are almost identical to those provided by the simple Durbin–Watson estimate of (2.3) or the MLE estimate from (2.11), underestimating $\rho$ when it approaches 1 with smaller $N$. In the case of errors due to a first-order AR process, taking all residual differences into account with a semivariogram appears to have no advantage over the much simpler Durbin–Watson technique, which only considers residual differences between nearest neighbours.

Accordingly, we generalize the concept of the sum of squares of forward differences from a one-dimensional time-line, to space of any number of dimensions, by constructing the *nearest new neighbour path* through space from data point to data point, starting with the point having the greatest sum of distances to all other data points. Then move from point to point, each time going to the nearest neighbour not already selected, until all data points are included in the path, as demonstrated in figure 7. Considering the OLS residuals at those points, we define the residual difference between each point and the next on the path to be the residual forward difference. In the case of equally spaced data points on a one-dimensional time-line, the sum of squares of these residual forward differences is the same SSRFD used to calculate the Durbin–Watson statistic in (2.3), so in the general case we use the term SSRFD for the sum of squares of the residual forward differences described here.

For equally spaced time-series data with $\rho > 0$, matrix $\mathbf{S}$ in (2.9) and (4.2) are identical if we set $r_0 = -t_s/\ln(\rho)$, where $t_s$ is the time between adjacent data points. In the general case of points unevenly distributed in space of one or more dimensions, we no longer have a uniform distance between the points along our path through space from point to point, so conventional methods described in §2, or our new methods described in §3, for specifying matrix $\mathbf{S}$ from $d = \text{SSRFD/SSR}$, may not necessarily be appropriate; this needs to be tested.

## 4.2. Random sampling null hypothesis testing

The techniques described above may be tested for sampling at random locations in space, for the null hypothesis of no actual trend, by first generating $N$ independent normally distributed samples, and then introducing correlation consistent with matrix $\mathbf{S}$ in (4.2), by pre-multiplying the vector of independent samples, by symmetric real matrx $\mathbf{S}^{0.5}$.

The magenta dashed and dotted lines in figure 8 demonstrate that OLS estimation, or assuming $r_0 = 0$, provides a completely unrealistic confidence test on trend or regression model significance, if the regression coefficient divided by its standard error is assumed to be $t$-distributed with $N - k - 1$ degrees of freedom, while GLS estimation with known $r_0$ is reasonably accurate. The same semivariogram method that worked well with the equally spaced data now performs poorly with the data randomly located in one-dimensional space.

The Durbin–Watson-based techniques of (2.3) and (3.1) perform well, if we define

$$\hat{r}_0 = -\frac{\bar{r}}{\ln(\hat{\rho})},$$

where

$\bar{r}$ is the mean distance between nearest new neighbours
$\hat{\rho}$ is estimated by (2.3) or (3.1).

(4.6)

Using this definition of $r_0$, it appears from comparison of figures 3 and 8, that the Durbin–Watson-based techniques of estimating correlation are insensitive to the non-uniform spacing between sample locations, while the semivariogram approach appears to be affected by the random spacing.

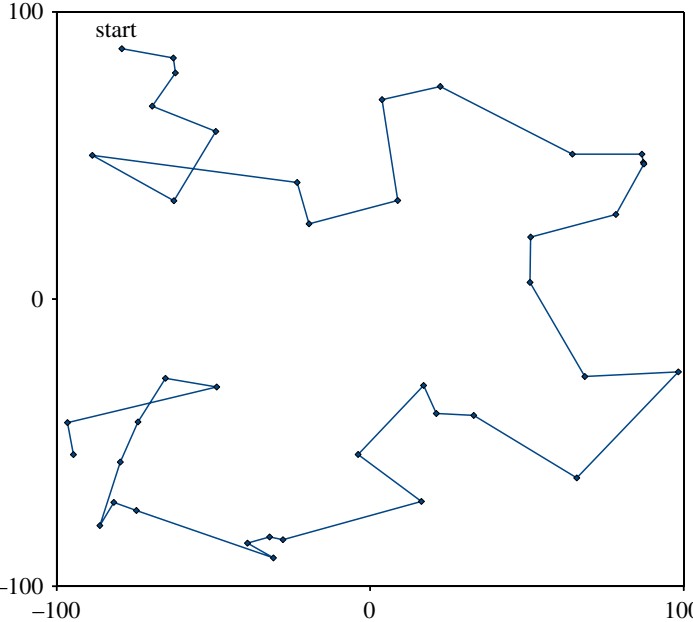

**Figure 7.** Nearest new neighbour path in two dimensions, with 40 randomly located points. The start point is the one with greatest sum of distances to all other points. At each step, the next one is the closest point not already selected. For locations with more than one data point, the nearest neighbour may be co-located, but to calculate the effective Durbin–Watson statistic, and mean distance $\bar{r}$ in (4.6), they are taken as a single point, with value the mean of the individual values.

## 4.3. Dealing with data points co-located in space or time

There may be more than one dataset available for a particular measured quantity, and they may each have measured values for the same time, or location in space.

If the datasets have similar residual variance $\sigma_e^2$, this may be handled by introducing distance $r_d$, assumed to be constant for all pairs of data points at the same time or location, from the two datasets. This $r_d$ represents the difference between datasets as equivalent to differences within the same dataset for that same distance. The value of the ratio $k_d = r_d/r_0$ may then be estimated from the $m$ pairs of residuals $e_i$ and $e_j$ that are at the same location in time or space as

$$k_d = \frac{r_d}{r_0} = -\ln\left[1 - \frac{1}{2m\sigma_e^2}\sum_{i=1}^{m}(e_i - e_j)^2\right]. \tag{4.7}$$

If dealing with a single dataset that has multiple measurements at some of the locations, then for all cases, except in calculating the diagonal elements of **S** where $i = j$, replace (4.1) with

$$\phi_{ij} = \exp\left[-\sqrt{\left(\frac{r_{ij}}{r_0}\right)^2 + k_d^2}\right]. \tag{4.8}$$

Alternatively, if combining two or more datasets, each with data at unique locations, but with the same locations in the datasets, use (4.1) when $i$ and $j$ are from the same dataset, but when $i$ and $j$ are from different datasets, use (4.8).

The semivariogram approach may still be used to fit the exponential function to the OLS residuals despite the $m$ co-located data points, but to estimate $\rho$ and hence $r_0$ from an equivalent to the Durbin–Watson described in §4.1, it is useful to construct a pre-multiplication matrix with $N$ columns and $N - m$ rows, which both sorts **X** and **e** in nearest new neighbour path order, and takes means of co-located rows. The resulting reduced vectors and matrices may be used in estimating $\rho$, the calculation of mean distance between nearest new neighbours $\bar{r}$, and the GLS validity testing of §2.6.

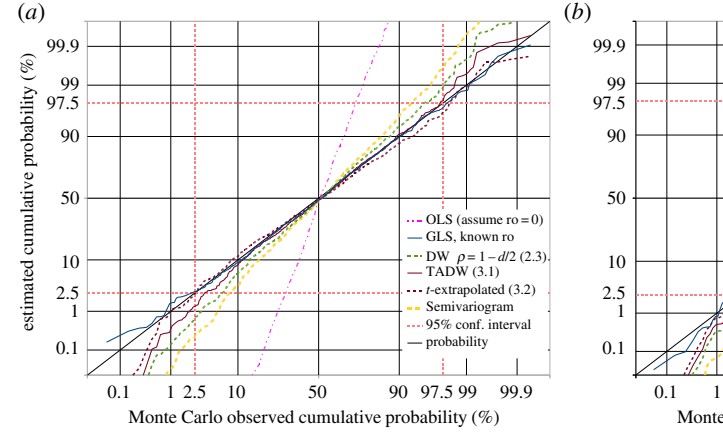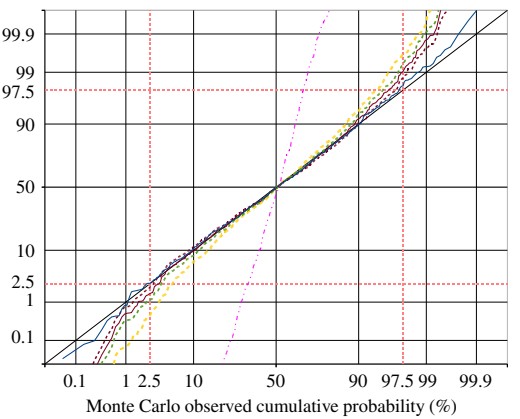

**Figure 8.** 1000 trials of GLS slope-$t$ estimation, with datasets of independent random samples at random locations over the range from $-100$ to $100$ in one dimension, correlated according to **S** given by (4.1) and (4.2) for known $r_0$. The $t$ estimates are compared with the $t$-distribution for $N - k - 1$ degrees of freedom (the diagonal black line). The dark blue solid line is the CDF of $t$-values produced using known $r_0$. Estimated $r_0$ values give poorer agreement: the semivariogram estimate (thick dashed yellow line) is poorest, the Durbin–Watson statistic (green dashed line) is better, and our improved Durbin–Watson estimate of $r_0$ (continuous dark brown line) is generally even better. Generally, the best is the dashed dark brown line, extrapolating from the two Durbin–Watson derived estimates. The dashed and dotted magenta line demonstrates the overestimate of trend significance if using OLS estimation. (*a*) $N = 100$, $r_0 = 11$, (*b*) $N = 300$, $r_0 = 6$.

## 4.4. Combining two datasets—global temperature anomaly data

To demonstrate the use of (4.7) and (4.8), our example is global monthly temperature data from two recently studied [16] surface climatological datasets, noting that NOAAGlobalTemp [17] provides temperature anomaly data from 1880 to the present, and HADCrut4 [18] provides similar data from 1850 to the present. For times after 1880, both datasets provide global temperature anomaly for the same times, creating pairs of points with zero time difference, if the two datasets are combined into one to obtain a combined model.

The annual means for the 120-year period 1897–2016 for the two datasets are shown in figure 9, together with quadratic best-fit curves and prediction intervals obtained by two different methods.

For both datasets, the Durbin–Watson test on OLS residuals of the monthly data fails, indicating GLS estimation may be required. However, in both cases, the Durbin–Watson test on the transformed residuals of the monthly data, described in §2.6, fails for GLS estimation assuming a first-order AR model. While this may be corrected with a higher-order model for the correlation, a simple alternative is to take annual means of the monthly data. Using the annual means, GLS estimation assuming a first-order AR model is successful.

The simple approach is to take the mean of the two values for each year, and treat it as an equally spaced time series. Fitting a quadratic model, OLS estimation results in a residual Durbin–Watson statistic of 0.802, so clearly either GLS estimation is required, or the quadratic model is not appropriate, or both. Assuming a first-order AR process, the estimates of $\rho$ from (2.2), (2.3), (2.11) and (3.1) are similar, yielding values of 0.582, 0.599, 0.592 and 0.637, respectively. All pass the Durbin–Watson test on the transformed residuals, with this test indicating (2.3) to be the best transformation, followed by (3.1), (2.11) and (2.2).

The other approach retains the 120 annual means from each dataset as separate data points, dealing with the co-located points as described above in §4.3. Again we have $\rho$ estimates from (2.3) and (3.1) of 0.599 and 0.637, respectively, and with $\bar{r} = 1$ year, we have $r_0 = 1.951$ years and $r_0 = 2.216$ years, respectively. Of these, the estimate from (2.3) again is indicated by the Durbin–Watson test on the transformed residuals to be the better, although both are acceptable with $d = 2.083$ and $d = 2.165$, respectively. The semivariogram estimate is $r_0 = 3.27$, which appears to be an overestimate, as it only just passes the transformed residual test with $d = 2.355$.

The best-fit curves from both approaches are almost identical, with positive slope of $+0.00798$ degrees per year for the period shown in figure 9, and curvature of $+0.00007$ degrees per year squared, and similar standard errors and prediction intervals. Both methods still indicate statistically significant positive slope and curvature of global temperature anomaly. Slope $t = 12.84$ for the mean

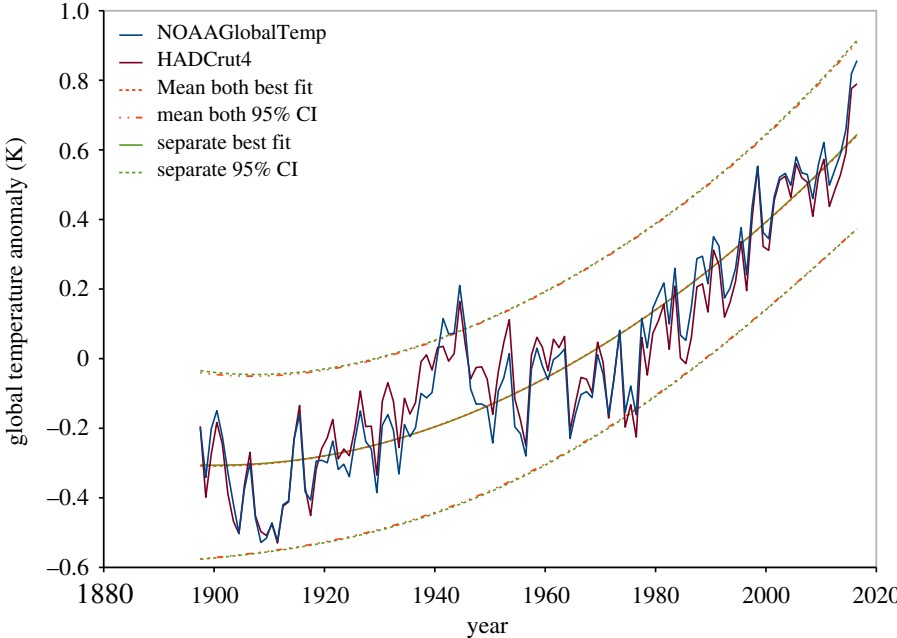

**Figure 9.** The two datasets of figure 5, NOAAGlobalTemp and HADCrut4, combined into one dataset, with GLS estimation, based on the simple Durbin–Watson estimate, $\rho = 1 - d/2$. Two approaches are used. One takes the mean of the two datasets for each year to combine into a single dataset (the red dashed best fit, and red dashed and dotted prediction interval), and uses conventional methods for equally spaced time-series data described in §2. The other (the green continuous best fit, and green dashed prediction interval) takes the 120 1-year means from each dataset, keeping the 240 observations separate in a dataset of 240 points, analysed as described in §4.3. The best-fit curves and the calculated prediction intervals are almost the same. However, the difference is barely visible here, due to the high correlation between the two datasets. Both methods indicate statistically significant positive slope and curvature of global temperature anomaly.

approach, and $t = 12.93$ for the separate approach, while curvature $t = 3.59$ for the mean approach, and $t = 3.64$ for the separate approach.

## 4.5. Summary of the proposed GLS, for data in multiple dimensions

The procedure developed above may be summarized as follows:

Initially perform OLS analysis of the data using the proposed model parameters. Then check for correlation between the residuals to test the validity of the OLS regression analysis. If the data is equally spaced in one dimension, this check is conventionally done by examining the ACF of the residuals, or their Durbin–Watson statistic, comparing with appropriate confidence limits. These are approximated in (2.7) for Durbin–Watson, as given in [6].

In the case of multiple dimensions, or unequally spaced data in one dimension, conventionally the semivariogram of residuals may be constructed. We propose a new alternative, that of finding the nearest new neighbour path connecting the data point locations, described in §4.1 above and figure 7. Order the residuals according to this path and calculate the effective Durbin–Watson statistic accordingly, and test against the confidence limits given in (2.7).

If these tests indicate correlation between residuals is unlikely, then assume that the OLS estimate and its $t$-test results on the significance of regression coefficients are reliable, and GLS estimation need not be undertaken.

If the Durbin–Watson statistic indicates spatial correlation is likely, a model for this correlation is required. Assuming an exponential model may be appropriate, we provide two to trial: the estimates of either (2.3) or (3.1), combined with (4.6) to estimate $\hat{r}_0$ in the model. Perform GLS analysis using these two options, and the Durbin–Watson test on transformed residuals as described in (2.12) and (2.13). The better of the two options from (2.3) and (3.1) is indicated by the better of the two Durbin–Watson test results. If both fall outside the required confidence limits, then conclude that our simple exponential model is unsuitable, and instead use conventional semivariogam analysis, together with a wider range of models [15].

(*a*)

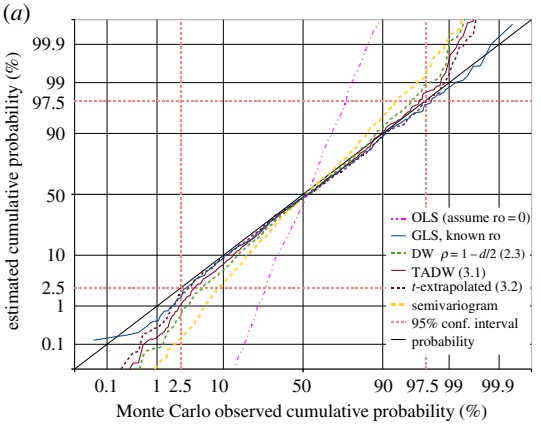

(*b*)

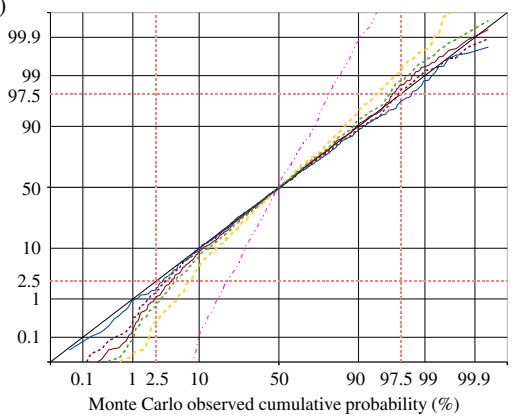

**Figure 10.** 1000 trials of GLS slope-$t$ estimation, with datasets of independent random samples at random locations over the range from $-100$ to $100$, in two- and three-dimensional space, correlated according to **S** given by (4.1) and (4.2) for known $r_0$. The $t$ estimates are compared with the $t$-distribution for $N - k - 1$ degrees of freedom (the diagonal black line). The dark blue solid line is the CDF of $t$-values produced using known $r_0$. While GLS using the semivariogram estimate of $r_0$ (the yellow dashed line) is a significant improvement over OLS (the magenta dashed and dotted line), the Durbin–Watson estimates using the nearest new neighbour path provide a $t$-statistic closer to that with known $r_0$. Closest is the extrapolation of (3.2) from the two Durbin–Watson estimates (dashed brown line). (*a*) $N = 100$ in 2D space, $r_0 = 35$, (*b*) $N = 100$ in 3D space, $r_0 = 50$.

If the two Durbin–Watson test results for the models from (2.3) and (3.1) are both within confidence limits, and both less than $E(d)$, with (3.1) closer than (2.3), then the $t$-value extrapolation (3.2) may be used. We suggest this criterion limiting its use, as real data may not behave the same as the ideal artificial data in our Monte Carlo testing in figures 3, 8 and 10, where (3.2) seemed generally a better option than (2.3) or (3.1).

There are a number of assumptions in the above procedure. The derivation of $E(d)$ and $V(d)$ in [6] was for OLS analysis of equally spaced one-dimensional data, but will be asymptotically correct in this more general case, so we assume these limits are reasonably accurate for small datasets in multiple dimensions. Another assumption of our method is that the spatial correlation is isotropic within the dimensions occupied by the data. Anisotropic semivariograms and their characterization are discussed in detail in [19]. Further detail on spatial statistical methods is provided in [20].

We assume that if the Durbin–Watson test on the pre-multiplied GLS residuals is satisfactory, then an exponential model is appropriate. If there is doubt about this last assumption, a useful test is to construct a semivariogram of the GLS pre-multiplied residuals **Pe** defined in (2.12).

We note that our method described here only tests for and corrects short-range dependence of the error process. If long-range dependence of the error process is suspected, either from the physical nature of the problem, or from long lag results in the ACF or semivariogram of residuals, then the simple procedure described in this paper may not be appropriate. A comparison of short and long memory models is given in [5].

# 5. Examples of GLS with regular and irregularly sampled data

## 5.1. Antarctic ice core temperature data

As an example of uniformly sampled time-series data, we analyse data at 1-year intervals between AD 1800 and AD 1999 from several ice core locations within central Antarctica [21]. The authors describe a modest warming over the last 150 years, based on this data combined with other evidence, so we take their reconstructed temperature data for 1850–1999, as our example.

A significant positive trend of 0.184 degrees per century is suggested by OLS estimation, with $t = 2.23$, but this apparent significance may not be valid, as the residual Durbin–Watson statistic is $d = 1.6$. The beta model of the distribution of $d$ indicates a probability of only 1% that $d$ would be that far removed from the uncorrelated expected value, $E(d) = 2.01$ in this case. Inspection of the temperature data, plotted in figure 11, suggests a linear trend model to be appropriate, so we conclude from $d = 1.6$ that there is modest positive error correlation, and try GLS estimation using a first-order AR model. This proves to be successful for all the estimates of $\rho$ described in this study, in that the

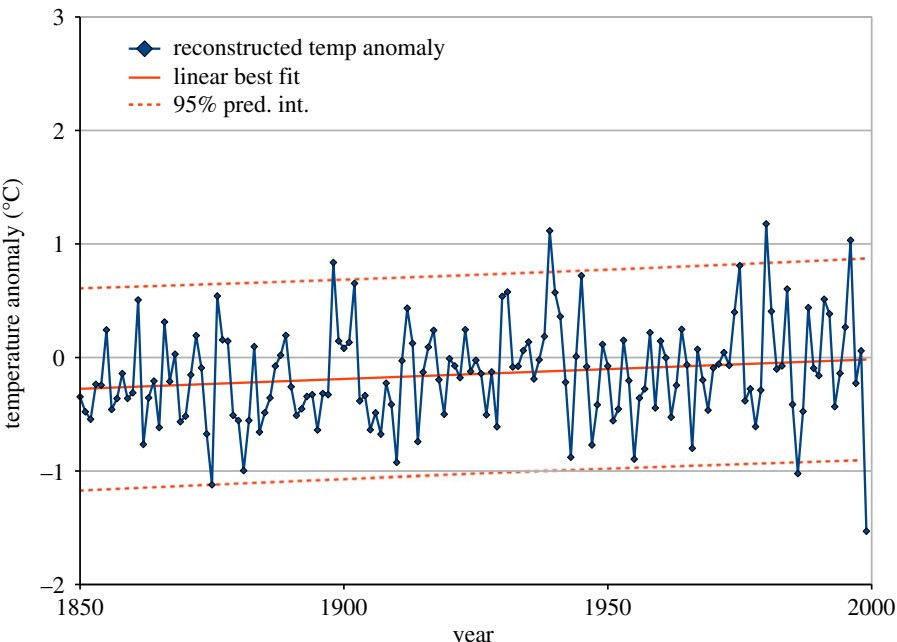

**Figure 11.** The last 150 years of annual data from a 200-year Antarctic ice core temperature record, with linear best fit and 95% prediction interval from GLS estimation. The estimate of $\rho = 0.216$ for the AR process is from (3.1). The trend is 0.018 degrees per century, which is not statistically significant. The 150-year result from Antarctic ice cores is in contrast to the 120-year result from the two worldwide datasets of figure 9, which has statistically significant slope of 0.8 degrees per century ($t = 12.8$) as well as upward curvature ($t = 3.6$).

Durbin–Watson statistic of the transformed GLS residuals does not differ significantly from the uncorrelated expected value.

The most successful estimate by this test is our new estimate (3.1), with $\rho = 0.216$ and transformed residual $d = 1.921$. The estimated trend with this GLS is 0.173 degrees per century, which is no longer significant at the 95% level. The simple estimate from (2.3), $\rho = 1 - d/2 = 0.200$ gives transformed GLS residual $d = 1.898$, ML while (2.11) and ACF lag 1 (2.1) both estimated $\rho = 0.159$ leading to an acceptable $d = 1.84$.

The danger of drawing conclusions from temperature trends observed over tens of years limited to the region of Antarctica has been described [21], and this caution is supported by our analysis of only this reconstructed temperature data. This result in figure 11 is very different from the trend seen in the two worldwide datasets shown in figure 9.

## 5.2. Cape Grim atmospheric methane concentration

We now take methane measurements at Cape Grim, figure 12, as an example of non-uniformly spaced data. These measurements [12] are available monthly from August 1984 onwards, but the first five measurements between April 1978 and 1984 were less frequent. This presents a problem for regression analysis, as the monthly measurements reveal a sinusoidal oscillation with a period of 1 year, as seen in the red line in figure 12. This leads to severe underestimation of $r_0$ using a simple polynomial model for the data, whether by our Durbin–Watson approach, or the conventional semivariogram method.

One potential solution to this problem is to add two parameters to the regression model for the data: the sine and cosine of $2\pi t$ where $t$ is the time in years. This provides an estimate of the phase and amplitude of the sinusoidal variation as well as the overall trend, if that is required. However, to fit just the overall trend, a simpler solution is to analyse annual means only, after the start of the monthly data. This approach has been used in the model fitting of figure 12.

The linear model from OLS estimation is shown in yellow, although a very low $d = 0.096$ indicates this is clearly not a valid result. The sparse measurements at the beginning of the series have greater influence on the GLS best-fit line. This is based on the estimate of $\rho = 0.978$ from our new model (3.1), leading to $r_0 = 47.84$ years. This very high apparent correlation may be due to an unsuitable

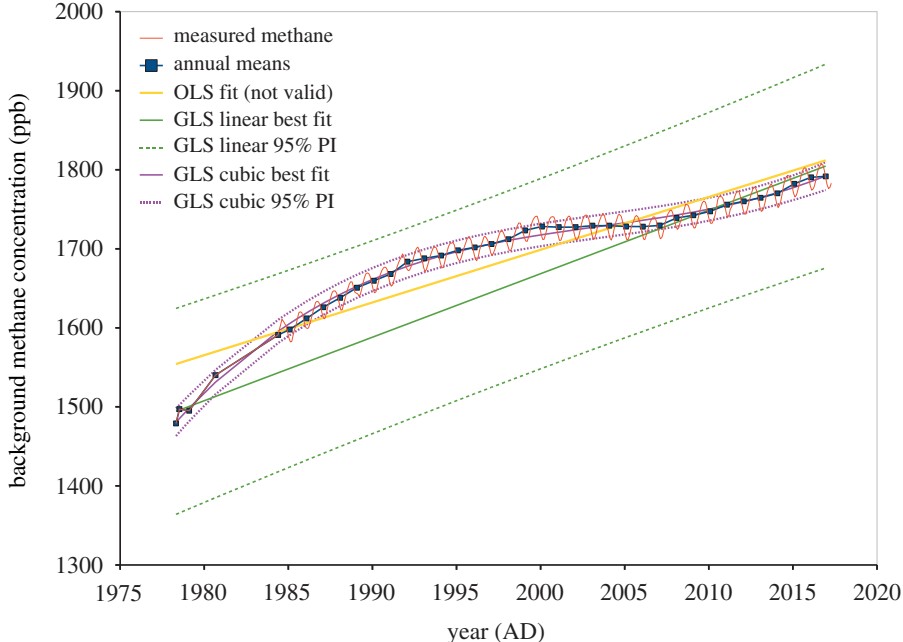

**Figure 12.** Cape Grim, Tasmania, background methane concentration measurements (thin red line), with annual sinusoidal variations evident after commencement of monthly measurements in mid-1984. This oscillation results in unacceptable values of the residual Durbin–Watson statistic $d$, so from this time on, annual means (dark blue line) are analysed instead. The best-fit line from OLS estimation is shown in yellow, although a very low $d = 0.096$ indicates this is clearly not a valid result. The sparse measurements at the beginning of the series have greater influence on the GLS best-fit line (green), in this case, based on the estimate of $\rho = 0.978$ and hence $r_0 = 47.84$ years from (3.1). Such high apparent correlation may suggest an unsuitable model, as does the excessive size of the prediction interval. A cubic model (magenta curve) has statistically significant regression parameters, lower residual standard error, and much more appropriate calculated prediction interval. The cubic model has $\rho = 0.451$ and hence $r_0 = 1.31$ years from (2.3).

model, as we saw earlier with the Cape Grim carbon dioxide data. In this case of methane data, the unsuitable linear model has the added disadvantage of overestimated errors, and an excessively large prediction interval.

A cubic model has regression parameters that are all statistically significant, and the estimate of correlation is much lower. Using (2.2) we have $\rho = 1 - d/2 = 0.451$, or $r_0 = 1.31$ years. The slope, curvature and cubic GLS regression coefficients are 3.88, $-0.172$, and 0.0105, respectively, with $t$-values 11.0, $-12.6$ and 8.3, respectively. The residual standard error is now 6.56 ppb, when compared with 45.3 ppb for the linear GLS model.

## 5.3. Radio climate modelling in two dimensions

As our final example, we examine the regression modelling of clear-air atmospheric effects on microwave radio propagation in the surface layer of the atmosphere. The dominant clear-air characteristics of the atmosphere affecting radio propagation are vertical moisture gradients and temperature gradients, as they produce vertical refractive index gradients, which cause the predominantly horizontally propagating radio signals to curve up or down. It is important to not only know the mean gradient over the surface layer; gradient variations within the surface layer may have a significant effect [22].

Ideally, we may imagine this problem being solved by numerical weather prediction (NWP) estimating the state of the lower atmosphere with sufficient resolution, of the order of 1 m vertically, and accuracy, to predict the radio propagation with a technique such as the parabolic equation model [23]. However NWP models do not yet have sufficient resolution, or accuracy in many locations [24], so empirical regression models are used. The current accepted prediction model [25] is an OLS estimate from a number of climatic and physical parameters [26]. Recently, an improved OLS regression model, taking advantage of new multipath fading data from countries not previously included in the dataset, and using additional parameters from surface weather station data, has been developed [27].

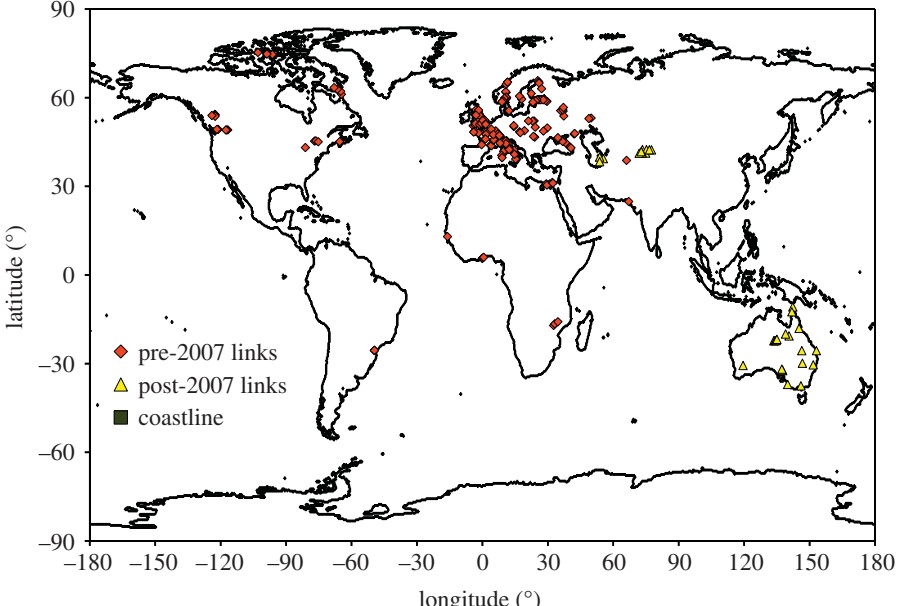

**Figure 13.** Radio links providing mutipath fading data points for the OLS models. Data collected prior to 2007 was included in development of the current ITU-R model [25], and the later data was added in developing the recent model [27].

These new parameters were calculated from weather station temperature, pressure, and dew-point measurements, and then interpolated for the radio link location by *natural neighbour interpolation* [28], using spherical geometry. This very effective technique [29] is perhaps under-used, as it is more difficult to code than kriging, yet it has some significant advantages: it only uses local data, does not require fitting of a semivariogram function to the data, does not assume stationarity of the data, yet provides smooth contours, and always provides stable exact interpolation, unlike kriging, which often may only provide an approximate fit, by employing the nugget effect [14], to remove the constraint of exactly fitting the data.

A significant motivation for this GLS study is that these fading prediction models were fitted to data from radio links around the world, with a very non-uniform distribution, seen in figure 13, and concern that the statistical significance attributed to regression coefficients may be overestimated.

The most significant predictor of fade depth $A_{0.01\%}$ was found [27] to be a composite parameter $v_1$, in terms of path length $D$ (km), mean rayline height at standard refractivity gradient $H_{8500}$, and $N_{sA90-10}$, the interdecile range of the time-series of surface refractivity anomaly $N_{sA}$ [24]:

$$v_1 = \frac{(N_{sA90-10}^{0.3})(D^{0.5})}{H_{8500}^{0.25}}. \tag{5.1}$$

Here, $N_{sA}$ is the difference between surface atmospheric radio refractive index, and its median value for the same hour of the day, and month of the year. The radio refractive index is estimated from atmospheric temperature, pressure, and water vapour pressure [30,31].

The correlation of this purely empirical $v_1$ parameter with fading, as a single-variable model, seemed extraordinary, with OLS estimation indicating $t = 18.94$. The GLS estimates described here are unanimous in relegating this parameter to $t = 10.0$ as a single-variable model. We now review the full nine-variable model [27].

Although OLS estimation was used in the initial study [27], the spatial Durbin–Watson test we introduce in this study indicates moderate but significant positive spatial correlation, with $d = 1.504$, so we repeat the regression analysis with GLS estimation. Results of this are summarized in table 1. The parameters in this table are described in detail in [27].

There are 327 available data points [27], but only 180 of them are at unique locations; the rest are at locations with more than one data point, due to differences in parameters such as radio frequency or antenna height. For this nine-variable model, we estimate $\rho = 1 - d/2 = 0.248$, resulting in $r_0 = 292.8$ km, and with $k_d = 0.3095$ we have an effective co-located distance $r_d = 90.6$ km. The $\rho = 1 - d/2$ estimate from (2.2) is used, as the resulting Durbin–Watson statistic of transformed GLS residuals is $d = 2.001$, which is slightly better than with (3.1): $r_0 = 319.5$ km and $d = 1.989$, or with the

**Table 1.** Nine-variable regression models of radio link multipath fading.

| parameter | OLS $\beta$ | $t$-value | DF | Pr($>$$|t|$) | GLS $\beta$ | $t$-value | DF | Pr($>$$|t|$) |
|---|---|---|---|---|---|---|---|---|
| (intercept) | $-20.8683$ | $-3.254$ | 317 | 0.001262 | $-26.7048$ | $-4.602$ | 317 | 0.000061 |
| $\log(1+|E_p|)$ | $-8.6974$ | $-7.823$ | 317 | 0.000000 | $-7.0148$ | $-6.073$ | 317 | 0.000000 |
| $\log(D)$ | 19.4386 | 7.258 | 317 | 0.000000 | 22.6975 | 7.307 | 317 | 0.000000 |
| $dN_1$ | $-0.0295$ | $-6.861$ | 317 | 0.000000 | $-0.0219$ | $-2.905$ | 317 | 0.003932 |
| $v_2$ | 0.0737 | 6.469 | 317 | 0.000000 | 0.1071 | 4.868 | 317 | 0.000002 |
| $dN010_{ERAI}$ | 0.0404 | 4.786 | 317 | 0.000003 | 0.0133 | 0.856 | 317 | 0.392886 |
| $NsA_{0.1\%}$ | $-0.1667$ | $-5.358$ | 317 | 0.000000 | $-0.0919$ | $-1.468$ | 317 | 0.143218 |
| $v_1$ | 1.6982 | 4.381 | 317 | 0.000016 | 1.4516 | 2.904 | 317 | 0.003941 |
| $H_L$ | $-0.0039$ | $-4.067$ | 317 | 0.000060 | $-0.0029$ | $-2.231$ | 317 | 0.026382 |
| $\log(f+6)$ | 7.9777 | 2.352 | 317 | 0.019272 | 8.0420 | 2.747 | 317 | 0.006356 |

semivariogram estimate $r_0 = 69$ km and $d = 1.997$. The larger $r_0$ values generally result in a more severe test of significance than low $r_0$.

One of the interesting results of the OLS regression is significant and roughly equal and opposite correlation of the two parameters $dN_1$ and $dN01_{ERAI}$, as they are both [25,32] predictions of the 1% point of the distribution of refractivity gradient in the 65 m surface layer of the atmosphere, by re-analysis of the NWP of the European Centre for Medium-range Weather Forecasting. The difference is that $dN_1$ is from an early re-analysis at 1.5 degrees latitude and longitude resolution, of 2 years of weather data, while $dN01_{ERAI}$ is from a much more recent and comprehensive re-analysis, at 0.75 degrees latitude and longitude resolution. Our GLS analysis now shows this arguably better version of the parameter to be insignificant in fading prediction, and the early version of the parameter remains significant, when used together with the other parameters in this regression model.

# 6. Conclusion

The statistical significance of OLS regression models may be overestimated when residuals are correlated with each other. Often this is a positive correlation with residuals that are nearby in space or time. Although there are existing GLS techniques for equally spaced time-series data to counteract this, they may not provide adequate correction in the case of highly correlated residuals and small datasets.

For equally spaced data, we describe a new estimate of the first-order autocorrelation, based on the Durbin–Watson statistic $d$, of the OLS residuals, and the exact expressions for expected value and variance of $d$. Monte Carlo testing shows this new non-iterative estimate to have similar variance to iterative ML estimation, but with significantly less bias for small positively correlated datasets. Further testing shows that this new model provides a valid statistical test of trend or regression model significance, using a standard $t$-test.

We generalize these techniques to randomly spaced data in the space of one or more dimensions, by the new concept of the nearest new neighbour path. This is compared with the standard technique of semivariogram estimation, and found to be superior in Monte Carlo testing. Tests of real data described here suggest the value of applying these techniques as well as semivariogram estimation, as we describe a validity test for the GLS scheme that may be applied to data randomly located in space.

Our focus in this study is on small datasets, so we considered only first-order AR models in the case of equally spaced data in one-dimensional space, and the equivalent exponential model in the general case. All the methods described here are asymptotically correct, so may be applied to large datasets; except that long-memory models of correlation may need to be considered. However, examples presented here suggest that for smaller datasets, choosing an appropriate model for the response data is often more important than the choice of model for the error correlation.

Data accessibility. Cape Grim data is at http://www.csiro.au/greenhouse-gases/. NOAAGlobaTemp data is at https:// www1.ncdc.noaa.gov/pub/data/noaaglobaltemp/operational/timeseries/ and HADCrut4 data is at http://www. metoffice.gov.uk/hadobs/hadcrut4/data/current/download.html. Detailed Antarctic ice core data for the last 200 years is at https://www1.ncdc.noaa.gov/pub/data/paleo/icecore/antarctica/antarctica-temp2006.txt. Radio

multipath fading model data, and code for the analysis of figures 1, 2, 6, 9 and 12 and table 1 is at https://github.com/radiosteve/OLSandGLS/.

Authors' contributions. S.J.S. performed the analysis; S.J.S. wrote the paper; D.A. and S.J.S. conceived the study; H.J.H. and D.A. supervised the study, and H.J.H. and D.A. proofed the manuscript.

Competing interests. No competing interests.

Funding. This work was supported by an Australian Government Research Training Scholarship.

Acknowledgements. The authors acknowledge the Australian Bureau of Meteorology, and CSIRO Oceans & Atmosphere, who jointly provided the Cape Grim atmospheric carbon dioxide and methane data. We are grateful to S. J. Tuke for useful discussions, and thank the anonymous reviewers for detailed comments leading to improvement of this manuscript.

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
