## [Reviewer comments · Royal Society Open Science]

Review History

RSOS-171493.R0 (Original submission)

Review form: Reviewer 1 (Hans Liljenström)

Is the manuscript scientifically sound in its present form?

Yes

Are the interpretations and conclusions justified by the results?

Yes

Is the language acceptable?

Yes

Is it clear how to access all supporting data?

Yes

Do you have any ethical concerns with this paper?

No

Have you any concerns about statistical analyses in this paper?

I do not feel qualified to assess the statistics

Recommendation?

Accept with minor revision (please list in comments)

Comments to the Author(s)

As climate related data are used to an increasing degree in modeling and predicting future climate change, depending on primarily an increase of greenhouse gases emitted to the atmosphere, a proper statistical data analysis is of paramount importance. Scientific results, in addition to informed interpretations, guide policy makers and provide a basis for societal decisions. This paper describes how the use of different statistical methods can give quite different results, showing trends of variable slopes, and points at the importance of being transparent in which method is being used and what it implies. This is important for the confidence in the conclusions made, and subsequently in the political (or other) decisions taken. The authors have developed a statistical method which seems to be more accurate than those commonly used, and thus should be of great significance for future studies. The new method also seems to be generally applicable, not only to climate related data, but to many other kinds of problems (e.g. biological), where there are unequally (or randomly) spaced observations in temporal or spatially distributed data sets.

While the statistical analysis appears to be correct and thoroughly worked out, the title of the paper is perhaps a bit misleading. The paper is more methodological than applied, and the climate related data used, is taken as an example and illustration to the differences between the statistical methods used. The authors should consider to change the title of the paper, to reflect this fact more than what the current title implies. Alternatively, that the paper is organized in another way, where the problems with trends in climatic data is initially more emphasized.

Review form: Reviewer 2

Is the manuscript scientifically sound in its present form?

No

Are the interpretations and conclusions justified by the results?

No

Is the language acceptable?

Yes

Is it clear how to access all supporting data?

Yes

Do you have any ethical concerns with this paper?

No

Have you any concerns about statistical analyses in this paper?

Yes

Recommendation?

Reject

Comments to the Author(s)

See the attached PDF file (Appendix A).

Decision letter (RSOS-171493.R0)

05-Jan-2018

Dear Mr Salamon:

Manuscript ID RSOS-171493 entitled "How real are observed climate trends?" which you submitted to Royal Society Open Science, has been reviewed. The comments from reviewers are included at the bottom of this letter.

In view of the criticisms of the reviewers, the manuscript has been rejected in its current form. However, a new manuscript may be submitted which takes into consideration these comments. It is imperative that a new manuscript on this topic make appropriate comparisons to other statistical methods, such as those described in detail by one of the reviewers. The suggestion of the other reviewer that the title of the manuscript itself be changed, in order to reflect a more methodological aspect of the work should also be taken on board.

Please note that resubmitting your manuscript does not guarantee eventual acceptance, and that your resubmission will be subject to peer review before a decision is made.

Your resubmitted manuscript should be submitted by 05-Jul-2018. If you are unable to submit by this date please contact the Editorial Office.

Please note that Royal Society Open Science will introduce article processing charges for all new submissions received from 1 January 2018. Charges will also apply to papers transferred to Royal Society Open Science from other Royal Society Publishing journals, as well as papers submitted as part of our collaboration with the Royal Society of Chemistry (<http://rsos.royalsocietypublishing.org/chemistry>). If your manuscript is submitted and accepted for publication after 1 Jan 2018, you will be asked to pay the article processing charge, unless you request a waiver and this is approved by Royal Society Publishing. You can find out more about the charges at <http://rsos.royalsocietypublishing.org/page/charges>. Should you have any queries, please contact openscience@royalsociety.org.

on behalf of Dr Andrew Wood (Associate Editor) and Jon Blundy (Subject Editor)
openscience@royalsociety.org

Reviewers' Comments to Author:

Reviewer: 1

Comments to the Author(s)

As climate related data are used to an increasing degree in modeling and predicting future climate change, depending on primarily an increase of greenhouse gases emitted to the atmosphere, a proper statistical data analysis is of paramount importance. Scientific results, in addition to informed interpretations, guide policy makers and provide a basis for societal decisions. This paper describes how the use of different statistical methods can give quite different results, showing trends of variable slopes, and points at the importance of being transparent in which method is being used and what it implies. This is important for the confidence in the conclusions made, and subsequently in the political (or other) decisions taken. The authors have developed a statistical method which seems to be more accurate than those commonly used, and thus should be of great significance for future studies. The new method also seems to be generally applicable, not only to climate related data, but to many other kinds of problems (e.g. biological), where there are unequally (or randomly) spaced observations in temporal or spatially distributed data sets.

While the statistical analysis appears to be correct and thoroughly worked out, the title of the paper is perhaps a bit misleading. The paper is more methodological than applied, and the climate related data used, is taken as an example and illustration to the differences between the statistical methods used. The authors should consider to change the title of the paper, to reflect this fact more than what the current title implies. Alternatively, that the paper is organized in another way, where the problems with trends in climatic data is initially more emphasized.

Reviewer: 2

Comments to the Author(s)

See the attached PDF file.

Author's Response to Decision Letter for (RSOS-171493.R0)

See Appendix B.

RSOS-181089.R0

Review form: Reviewer 1 (Hans Liljenström)

Is the manuscript scientifically sound in its present form?

Yes

Are the interpretations and conclusions justified by the results?

Yes

Is the language acceptable?

Yes

Is it clear how to access all supporting data?

Yes

Do you have any ethical concerns with this paper?

No

Have you any concerns about statistical analyses in this paper?

I do not feel qualified to assess the statistics

Recommendation?

Accept as is

Comments to the Author(s)

The authors have considered my comments satisfactory and changed the title accordingly.

Review form: Reviewer 3

Is the manuscript scientifically sound in its present form?

Yes

Are the interpretations and conclusions justified by the results?

No

Is the language acceptable?

Yes

Is it clear how to access all supporting data?

Yes

Do you have any ethical concerns with this paper?

No

Have you any concerns about statistical analyses in this paper?

No

Recommendation?

Major revision is needed (please make suggestions in comments)

Comments to the Author(s)

Manuscript ID RSOS-181089

Journal: Royal Society Open Science

How real are observed trends in small correlated datasets?

Authors: Salamon, Stephen; University of Adelaide Faculty of Engineering Computer and Mathematical Sciences,

Hansen, Hedley; University of Adelaide Faculty of Engineering Computer and Mathematical Sciences

Abbott, Derek; University of Adelaide, School of Electrical & Electronic Engineering

In this paper a new correlation estimate based on the Durbin-Watson statistic is developed, leading to an improved estimate of autoregression with highly correlated data, thus reducing this risk. These techniques are generalised to randomly located data points in space, through the new concept of the nearest new neighbour path. The authors describe tests on the validity of the GLS schemes, allowing verification of the models employed.

Although the paper could furnish a good contribution to the existing literature, however some significant aspects must be clarified and improved; without these specific requirements the paper cannot be accepted for publication. In particular:

- 1) It is not specified throughout the paper what are the theoretical hypotheses to apply the new correlation estimate, neither are specified the practical aspects outcoming for applying this new estimate. The above discussion concerns equally spaced data points in one dimensional space.
- 2) It is not clear how to apply the GLS estimate in a multi-dimensional space. Moreover and most importantly, what is its usefulness in this context? The authors suggest to replace a first order autoregressive model with its equivalent exponential model: however, this is very restrictive. Indeed, geostatistical literature provides several tools and better solutions in this context; for example, anisotropy aspects are not considered by the authors, moreover spatial correlation is well captured by the variogram and several models exist to describe the spatial (or, more generally, spatio-temporal) correlation.
- 3) At least in the spatial context, literature must be updated.

Then, in the present form the paper cannot be accepted. A major revisioni is required.

Decision letter (RSOS-181089.R0)

17-Dec-2018

Dear Mr Salamon,

The Subject Editor assigned to your paper ("How real are observed trends in small correlated datasets?") has now received comments from reviewers. We would like you to revise your paper in accordance with the referee and Associate Editor suggestions which can be found below (not including confidential reports to the Editor). Please note this decision does not guarantee eventual acceptance.

Please submit a copy of your revised paper before 09-Jan-2019. Please note that the revision deadline will expire at 00.00am on this date. If we do not hear from you within this time then it will be assumed that the paper has been withdrawn. In exceptional circumstances, extensions may be possible if agreed with the Editorial Office in advance. We do not allow multiple rounds of revision so we urge you to make every effort to fully address all of the comments at this stage. If deemed necessary by the Editors, your manuscript will be sent back to one or more of the original reviewers for assessment. If the original reviewers are not available we may invite new reviewers.

When submitting your revised manuscript, you must respond to the comments made by the referees and upload a file "Response to Referees" in "Section 6 - File Upload". Please use this to

document how you have responded to each of the comments, and the adjustments you have made. In order to expedite the processing of the revised manuscript, please be as specific as possible in your response.

- Ethics statement

- Data accessibility

If you wish to submit your supporting data or code to Dryad (<http://datadryad.org/>), or modify your current submission to dryad, please use the following link:
<http://datadryad.org/submit?journalID=RSOS&manu=RSOS-181089>

- Competing interests

- Authors' contributions

- Acknowledgements

- Funding statement

on behalf of Prof Jon Blundy (Subject Editor)
openscience@royalsociety.org

Editor Comments to Author:

Thank you for submitting this resubmission. After further review, the referees have recommended a major revision of the manuscript prior to reconsidering it. Please ensure that you fully respond to their feedback (by incorporating their recommendations and providing a full scientific rebuttal if you choose not to include aspects of their commentary). If the referees remain unsatisfied by the revision, we may not be able to consider the manuscript further. Good luck and we look forward to receiving the revision soon.

Reviewer comments to Author:

Reviewer: 1

Comments to the Author(s)

The authors have considered my comments satisfactory and changed the title accordingly.

Reviewer: 3

Comments to the Author(s)

Manuscript ID RSOS-181089

Journal: Royal Society Open Science

How real are observed trends in small correlated datasets?

Authors: Salamon, Stephen; University of Adelaide Faculty of Engineering Computer and Mathematical Sciences,

Hansen, Hedley; University of Adelaide Faculty of Engineering Computer and Mathematical Sciences

Abbott, Derek; University of Adelaide, School of Electrical & Electronic Engineering

In this paper a new correlation estimate based on the Durbin-Watson statistic is developed, leading to an improved estimate of autoregression with highly correlated data, thus reducing this risk. These techniques are generalised to randomly located data points in space, through the new concept of the nearest new neighbour path. The authors describe tests on the validity of the GLS schemes, allowing verification of the models employed.

Although the paper could furnish a good contribution to the existing literature, however some significant aspects must be clarified and improved; without these specific requirements the paper cannot be accepted for publication. In particular:

- 1) It is not specified throughout the paper what are the theoretical hypotheses to apply the new correlation estimate, neither are specified the practical aspects outcoming for applying this new estimate. The above discussion concerns equally spaced data points in one dimensional space.
- 2) It is not clear how to apply the GLS estimate in a multi-dimensional space. Moreover and most importantly, what is its usefulness in this context? The authors suggest to replace a first order autoregressive model with its equivalent exponential model: however, this is very restrictive. Indeed, geostatistical literature provides several tools and better solutions in this context; for example, anisotropy aspects are not considered by the authors, moreover spatial correlation is well captured by the variogram and several models exist to describe the spatial (or, more generally, spatio-temporal) correlation.
- 3) At least in the spatial context, literature must be updated.

Then, in the present form the paper cannot be accepted. A major revisioni is required.

Author's Response to Decision Letter for (RSOS-181089.R0)

See Appendix C.

RSOS-181089.R1 (Revision)

Review form: Reviewer 3

Is the manuscript scientifically sound in its present form?

Yes

Are the interpretations and conclusions justified by the results?

Yes

Is the language acceptable?

Yes

Is it clear how to access all supporting data?

Yes

Do you have any ethical concerns with this paper?

No

Have you any concerns about statistical analyses in this paper?

No

Recommendation?

Accept as is

Comments to the Author(s)

No comments. The paper is ok after reviewing

Decision letter (RSOS-181089.R1)

12-Feb-2019

Dear Mr Salamon,

I am pleased to inform you that your manuscript entitled "How real are observed trends in small correlated datasets?" is now accepted for publication in Royal Society Open Science.

on behalf of Professor Jon Blundy (Subject Editor)
openscience@royalsociety.org

Reviewer comments to Author:
Reviewer: 3

Comments to the Author(s)
No comments. The paper is ok after reviewing

Follow Royal Society Publishing on Twitter: [@RSocPublishing](https://twitter.com/RSocPublishing)
Follow Royal Society Publishing on Facebook:
<https://www.facebook.com/RoyalSocietyPublishing.FanPage/>
Read Royal Society Publishing's blog: <https://blogs.royalsociety.org/publishing/>

Appendix A

Review of ‘How real are observed climate trends?’

by S. J. Salamon, H. J. Hansen and D. Abbott

A main focus of this manuscript is the assessment of the significance of a trend in a time series when the series is assumed to be the sum of two components: a trend that can be expressed as a linear combination of explanatory variables (with one parameter for each variable) and normally distributed errors that obey a first-order autoregressive (AR(1)) process with an autocorrelation structure dictated by an additional parameter ρ . If ρ were known in this setup, its value could be used to estimate the trend parameters using generalized least squares (GLS), and the resulting estimated trend would be optimal in a certain minimum variance sense. If ρ is unknown, optimality can be recovered by using a maximum likelihood (ML) procedure to jointly estimate ρ and the trend parameters. This approach is well known and is addressed in standard textbooks on time series analysis under the subject of regression with stationary errors – see, for example, Section 6.6.2 of *Introduction to Time Series and Forecasting* (Third Edition) by P. J. Brockwell and R. A. Davis, Springer, 2016, or Section 3.8 of *Time Series Analysis and Its Applications: With R Examples* (Fourth Edition) by R. H. Shumway and D. S. Stoffer, Springer, 2017. Rather than using this standard approach to address the question posed by the title of the manuscript, the manuscript ignores it almost completely (item [5] in the detailed comments below explains the qualifier ‘almost’). Instead the manuscript considers a procedure in which ordinary least squares (OLS) is used to estimate the trend parameters, and the residuals from the fitted model are used to estimate ρ . This estimate in turn is then used to reestimate the trend parameters. The need to use an estimate for ρ causes estimates of the trend parameters to be suboptimal. With particular emphasis on the case where $|\rho|$ is close to unity, the authors put forth methodology evidently intended to overcome sub-optimality. Whereas the standard approach falls out from well-established statistical principles (ML estimation), the approach advocated in this manuscript is *ad hoc* in nature (see the detailed comments below). In addition, the standard approach has flexibility in the choice of a model for the errors, whereas the approach advocated here is based on the often – but not always – reasonable contention that an AR(1) process is a good choice for modeling correlated errors (or a well-known extension to this process for unequally spaced time series or for spatial data). The standard approach also advocates a careful study of the residuals from the final model to confirm its reasonableness, the lack of which in the current manuscript is a serious flaw. The lack of any serious comparison of the proposed the *ad hoc* methodology with the optimal standard approach is another serious flaw.

Here are some detailed comments (listed roughly in the order in which the issues arise while reading through the manuscript).

- [1] The exposition surrounding Equation (2.1) employs the standard distinction in the statistical literature between errors (unobservable terms in a model) and residuals (observable terms resulting from fitting a model). This distinction is blurred elsewhere in the paper. Even if errors are assumed to be uncorrelated, residuals from least squares fits are not. This fact is discussed in standard references on regression analysis (e.g., *Applied Linear Regression* (Fourth Edition) by S. Weisberg, Wiley, 2014), but it follows readily from the fact the sum of least squares residuals must be zero. The first sentence in Section (b) says ‘... if the residuals are correlated, a better estimator may

be found’, but the statement applies to errors, not residuals (there are other instances in the manuscript where a similar clarification is needed).

- [2] While an AR(1) model is often adequate for modeling noise in climate time series, its structure for dependence is short range in the sense of exhibiting exponential decay as the time between variates increases. Models encapsulating long-range dependence (hyperbolic as opposed to exponential decay) have been gaining in popularity. As formulated using a fractional difference process or fractional Gaussian noise, these models are as simple as an AR(1) model in the sense of having the same number of unknown parameters (two). A good entrance into the literature that discusses the relative merits of models with short- and long-range dependence is the article ‘Reconstructing Past Temperatures from Natural Proxies and Estimated Climate Forcings using Short- and Long-Memory Models’ by Barboza et al., *Annals of Applied Statistics*, Vol. 8, No. 4, 2014, pp. 1966–2001. As noted previously, the current manuscript only delves into AR(1) noise, which is too restrictive to provide an authoritative answer to the sweeping question posed in the title of the manuscript.
- [3] In the expression for r_k in Equation (2.2), ‘ $i = 2$ ’ should be ‘ $i = 1$ ’ to get the standard expression for the sample ACF.
- [4] The final sentence in the paragraph containing Equation (2.9) doesn’t make sense to me – given that OLS is easier to implement than GLS, wouldn’t the criterion of simplicity suggest OLS to be a preferred approach over GLS?
- [5] In the next paragraph, the Cochrane–Olcutt method is briefly discussed, but nothing more is done with it in the rest of the paper. Both Brockwell & Davis and Shumway & Stoffer discuss the role that the Cochrane–Olcutt method can play in obtaining the optimal joint ML estimates of ρ and the trend parameters. This is the only indication in the manuscript of an awareness of the optimal standard approach.
- [6] I fail to see why the failure of the error bars in Figure 1 to overlap ‘only a few years either side of the measured data’ tells us much about the standard errors deduced for slope and curvature, as suggested toward the end of Section 2(d). I also fail to see why the greater degree of overlap in Figure 4 in any way validates the *ad hoc* approach advocated in this manuscript. Trend is an inherently ill-defined concept that only becomes well defined with an assumption about the nature of the trend. I see no reason why the assumptions of linear and quadratic trends should lead to trend extrapolations that agree with one another.
- [7] I fail to see why requiring the t -test for the regression slope to obey a t -distribution with $N - k$ degrees of freedom should be regarded as a fundamental requirement (here $k = 1$ for a linear trend, while $k = 2$ for a quadratic trend). There are two additional parameters (ρ and an intercept parameter) for a total of $k + 2$ parameters. In view of the optimal standard approach, is there any relevant statistical theory suggesting that a t -distribution with $N - k$ degrees of freedom is relevant for assessing the significance of an optimally estimated slope parameter? Without supporting statistical theory and without direct comparison to the optimal estimator, the empirically determined estimator $\hat{\rho}$ of Equation (3.2) is of questionable value.
- [8] In Figure 3, are stationary initial conditions used to initiate the simulation of AR(1) noise? Use of such conditions is important to use for values of ρ close to unity.

- [9] In the discussion following Equation (4.2), it is not clear to me why inversion of the matrix $\mathbf{X}'\mathbf{S}^{-1}\mathbf{X}$ is regarded as a likely problem. The dimension of this matrix is $(k+1) \times (k+1)$, which is quite small when k is either 1 or 2, as is assumed throughout the manuscript.
- [10] There is a substantial literature on using the semivariogram to estimate model parameters, none of which is mentioned in the current manuscript. Two references that come to mind are *Interpolation of Spatial Data* by M. L. Stein, Springer–Verlag, 1999, and *Geostatistics: Modeling Spatial Uncertainty* (Second Edition) by J.-P. Chilès and P. Delfiner, Wiley, 2012 (use of these with a citation index should lead to current thinking on semivariogram-based estimators).

Appendix B

Manuscript ID: RSOS-171493

Type: Regular

Title: "How real are observed climate trends?"

Revised title: "How real are observed trends in small correlated datasets?"

Author: Salamon et al.

To: Alice Power, Editorial Coordinator

Dear Alice,

Re: Reply to reviewers on MS # RSOS-171493

Many thanks for having our paper RSOS-171493 reviewed.

We have pleasure in attaching an updated manuscript and our point-by-point response to the comments is given below.

Best regards,

Stephen Salamon et al.

Reviewer#1: As climate related data are used to an increasing degree in modeling and predicting future climate change, depending on primarily an increase of greenhouse gases emitted to the atmosphere, a proper statistical data analysis is of paramount importance. Scientific results, in addition to informed interpretations, guide policy makers and provide a basis for societal decisions. This paper describes how the use of different statistical methods can give quite different results, showing trends of variable slopes, and points at the importance of being transparent in which method is being used and what it implies. This is important for the confidence in the conclusions made, and subsequently in the political (or other) decisions taken. The authors have developed a statistical method which seems to be more accurate than those commonly used, and thus should be of great significance for future studies. The new method also seems to be generally applicable, not only to climate related data, but to many other kinds of problems (e.g. biological), where there are unequally (or randomly) spaced observations in temporal or spatially distributed data sets.

While the statistical analysis appears to be correct and thoroughly worked out, the title of the paper is perhaps a bit misleading. The paper is more methodological than applied, and the climate related data used, is taken as an example and illustration to the differences between the statistical methods used. The authors should consider to change the title of the paper, to reflect this fact more than what the current title implies. Alternatively, that the paper is organized in another way, where the problems with trends in climatic data is initially more emphasized.

Author reply: We thank the reviewer for these comments. We agree a different title would better reflect the nature of the paper. We also agree the method has more general applications.

Author action: We have changed the title of the paper to "How real are observed trends in small correlated datasets?" To emphasize generality, we have added the following to the abstract: "The method is generally applicable, not only to climate related data, but to many other kinds of problems (eg. biological and medical data), where there are unequally (or randomly) spaced observations in temporal or spatially distributed data sets."

Reviewer#2: A main focus of this manuscript is the assessment of the significance of a trend in a time series when the series is assumed to be the sum of two components: a trend that can be expressed as a linear combination of explanatory variables (with one parameter for each variable) and normally distributed errors that obey a first-order autoregressive (AR(1)) process with an autocorrelation structure dictated by an additional parameter ρ . If ρ were known in this setup, its value could be used to estimate the trend parameters using generalized least squares (GLS), and the resulting estimated trend would be optimal in a certain minimum variance sense. If ρ is unknown, optimality can be recovered by using a maximum likelihood (ML) procedure to jointly estimate ρ and the trend parameters. This approach is well known and is addressed in standard textbooks on time series analysis under the subject of regression with stationary errors – see, for example, Section 6.6.2 of *Introduction to Time Series and Forecasting (Third Edition)* by P. J. Brockwell and R. A. Davis, Springer, 2016, or Section 3.8 of *Time Series Analysis and Its Applications: With R Examples (Fourth Edition)* by R. H. Shumway and D. S. Stoffer, Springer, 2017.

Rather than using this standard approach to address the question posed by the title of the manuscript, the manuscript ignores it almost completely (item [5] in the detailed comments below explains the qualifier 'almost'). Instead the manuscript considers a procedure in which ordinary least squares (OLS) is used to estimate the trend parameters, and the residuals from the fitted model are used to estimate ρ . This estimate in turn is then used to re-estimate the trend parameters. The need to use an estimate for ρ causes estimates of the trend parameters to be suboptimal. With particular emphasis on the case where $|\rho|$ is close to unity, the authors put forth methodology evidently intended to overcome sub-optimality. Whereas the standard approach falls out from well-established statistical principles (ML estimation), the approach advocated in this manuscript is ad hoc in nature (see the detailed comments below).

Author reply: We thank the reviewer for a most detailed examination of the methodology of this paper. We agree that maximum likelihood (ML) estimation should have been discussed in the paper. While ML is rightly considered optimal in the asymptotic sense, we would point out that ML estimation is not always unbiased, a potential problem when estimating ρ values close to unity. It is perhaps surprising that in all our testing with both simulated and real data, the results with iterated joint ML estimation of ρ and β , were almost indistinguishable from the far simpler technique described in the original manuscript, estimating ρ as $1 - d/2$, where d is the Durbin-Watson statistic from the OLS residuals; that is, without any iteration! We agree that the new model estimating of ρ in the original paper was ad-hoc, so we have now replaced it with an improved method.

Author action: We now include ML estimation in our background section 2 and include joint ML estimation of ρ and β alongside other techniques in all testing of equally-spaced time-series data in the paper. We have replaced the ad-hoc modified Durbin-Watson statistic of the original manuscript by a new estimate of ρ , this time based on the exact expressions published by Durbin and Watson for the expected value and variance of d for the null hypothesis case. We show that this new estimate has much lower bias than ML estimation for small datasets and ρ approaching unity, while achieving similar variance.

Reviewer#2: In addition, the standard approach has flexibility in the choice of a model for the errors, whereas the approach advocated here is based on the often – but not always – reasonable contention that an AR(1) process is a good choice for modeling correlated errors (or a well-known extension to this process for unequally spaced time

series or for spatial data). The standard approach also advocates a careful study of the residuals from the final model to confirm its reasonableness, the lack of which in the current manuscript is a serious flaw. The lack of any serious comparison of the proposed the ad hoc methodology with the optimal standard approach is another serious flaw.

Author reply: We agree that choice in the model for the errors is desirable, particularly in large datasets. However, our focus is on small positively correlated datasets, where we generally find flexibility in the choice of the model for the response is generally a more important issue, and this flexibility is not restricted in any way by our methodology. However, we now describe testing of the transformed GLS residuals, so that suitability of the AR(1) or exponential process may be tested. Where this testing fails despite trying different models for the response, different types of models may need to be considered for the error correlation, but that is beyond the scope of this paper. We provide the tools for the reader to recognize that situation.

Author action: A detailed description of a Durbin-Watson test on transformed GLS residuals, including calculation of the associated confidence interval, is now provided in section 2(f).

Reviewer#2 [1]: The exposition surrounding Equation (2.1) employs the standard distinction in the statistical literature between errors (unobservable terms in a model) and residuals (observable terms resulting from fitting a model). This distinction is blurred elsewhere in the paper. Even if errors are assumed to be uncorrelated, residuals from least squares fits are not. This fact is discussed in standard references on regression analysis (e.g., *Applied Linear Regression (Fourth Edition)* by S. Weisberg, Wiley, 2014), but it follows readily from the fact the sum of least squares residuals must be zero. The first sentence in Section (b) says `... if the residuals are correlated, a better estimator may be found', but the statement applies to errors, not residuals (there are other instances in the manuscript where a similar clarification is needed).

Author reply: We agree that the important distinction between errors and residuals, especially important for small sample sizes, was not sufficiently clear in the original manuscript.

Author action: The text in section 2(b) referred to above has now been changed to read "If the residuals are correlated, this suggests correlation between the errors, although the two are not the same, as bOLS is only an estimate of β . If the errors are correlated, a better estimator may be found, if a reasonable model for this correlation is available".

Reviewer#2 [2]: While an AR(1) model is often adequate for modeling noise in climate time series, its structure for dependence is short range in the sense of exhibiting exponential decay as the time between variates increases. Models encapsulating long-range dependence (hyperbolic as opposed to exponential decay) have been gaining in popularity. As formulated using a fractional difference process or fractional Gaussian noise, these models are as simple as an AR(1) model in the sense of having the same number of unknown parameters (two). A good entrance into the literature that discusses the relative merits of models with short- and long-range dependence is the article 'Reconstructing Past Temperatures from Natural Proxies and Estimated Climate Forcings using Short- and Long-Memory Models' by Barboza et al., *Annals of Applied Statistics*, Vol. 8, No. 4, 2014, pp. 1966–2001. As noted previously, the current manuscript only delves into AR(1) noise, which is too restrictive to provide an authoritative answer to the sweeping question posed in the title of the manuscript.

Author reply: We agree that the sweeping question posed in the title tends to misrepresent the true nature of this paper. Our main focus is the difficult problem of

analyzing trends in relatively small and potentially highly positively correlated datasets. While including models with long-range dependence would have made the paper more general, for the sake of simplicity we only considered exponential decay models, as most relevant to small datasets. We have generalized the techniques described in terms of the number of dimensions of space where the observations are made, and whether they are uniformly or randomly located in that space, and even described how to deal with cases where some observations are co-located. However, long-range dependence is beyond the scope of this present study, as it is less likely to be encountered in small datasets.

Author action: Testing of the transformed GLS residuals is now described, so that suitability of the AR(1) or exponential process that we use may be tested. We have changed the title of the paper to "How real are observed trends in small correlated datasets?" to clarify the scope of the paper.

Reviewer#2 [3]: In the expression for r_k in Equation (2.2), ' $i = 2$ ' should be ' $i = 1$ ' to get the standard expression for the sample ACF.

Author reply: We agree and thank the reviewer for noticing this typographic error in (2.2). This error was only in the text, not the computer code used in the analysis.

Author action: The typographic error has been corrected.

Reviewer#2 [4]: The final sentence in the paragraph containing Equation (2.9) doesn't make sense to me – given that OLS is easier to implement than GLS, wouldn't the criterion of simplicity suggest OLS to be a preferred approach over GLS?

Author reply: Although simple scaling of OLS results appears to be an effective method of correcting the t-statistic according to ρ , it does not correct the estimated bOLS model according to ρ , while GLS does. The scaled OLS technique does not provide any way of testing that the assumed AR(1) model is appropriate. We now include a description of that testing for GLS estimation in 2(f).

Author action: We have replaced this sentence with: "A weakness of the schemes of (2.14) and (2.15) [the scaled OLS approach] are that they do not allow for testing of the correctness of the error correlation model, as described in section (f) above."

Reviewer#2 [5]: To In the next paragraph, the Cochrane–Olcutt method is briefly discussed, but nothing more is done with it in the rest of the paper. Both Brockwell & Davis and Shumway & Stoffer discuss the role that the Cochrane–Olcutt method can play in obtaining the optimal joint ML estimates of ρ and the trend parameters. This is the only indication in the manuscript of an awareness of the optimal standard approach.

Author reply: The Cochrane–Orcutt method was our initial focus in this research, given that for equally spaced time-series data with AR(1) noise of known ρ , Cochrane–Orcutt exactly removes the correlation in the errors. This comes with the disadvantage of the loss of one data point, which may be significant for small data-sets, which is the particular focus of this paper. This disadvantage may be overcome by Prais-Winsten estimation, which we did not mention, because it is essentially equivalent to GLS estimation, as described in our paper, and as implemented in R, and GLS is both a more familiar standard method, and more readily generalized to unevenly spaced data in multiple dimensions. We should have mentioned ML estimation – we have now corrected that oversight.

Author action: We now have a new sub-section 2(e) "Maximum likelihood estimation," where we describe a practical approach to this, based on Brockwell & Davis and Shumway & Stoffer, which we have successfully tested and found to be very stable. We use Shumway & Stoffer's estimate of ρ conditioned on the first error term, together

with the Brockwell & Davis iterative scheme to converge on the joint ML estimate. Interestingly, this ML method gives strikingly similar results to our far simpler use of the Durbin-Watson statistic to estimate ρ , as $1 - d/2$, described in our original manuscript.

Reviewer#2 [6]: I fail to see why the failure of the error bars in Figure 1 to overlap 'only a few years either side of the measured data' tells us much about the standard errors deduced for slope and curvature, as suggested toward the end of Section 2(d). I also fail to see why the greater degree of overlap in Figure 4 in any way validates the ad hoc approach advocated in this manuscript. Trend is an inherently ill-defined concept that only becomes well defined with an assumption about the nature of the trend. I see no reason why the assumptions of linear and quadratic trends should lead to trend extrapolations that agree with one another.

Author reply: We agree that this part of the original manuscript was not well explained. Mathematically there is no reason to expect the error bars for extrapolations of alternative models to overlap, but we have now clarified this in terms of prediction intervals, as defined in texts such as Probability & Statistics for Engineers & Scientists by Walpole, Myers, Myers, and Ye (9th ed., Prentice Hall). In practice, regression models may be called on to provide an estimate in a region where no data is available, and no acceptable physical model is available either. Our radio climate modelling example of sub-section 5(c) is an example of this. While such extrapolations may be undesirable, they are sometimes unavoidable, and it would be desirable in such circumstances that estimated confidence intervals from alternative models overlap the unknown real values – impossible if those intervals have no overlap with each-other. This was the reason for our statement that failure of the error bars to overlap beyond the available data casts doubt on the certainty implied by high t-values.

Author action: We now include a new section 2(i), "The prediction interval - supervised machine learning," which defines the prediction interval for GLS regression, and discusses with the example of figure 2, how it may be used for forecasting.

Reviewer#2 [7]: I fail to see why requiring the t-test for the regression slope to obey a t-distribution with $N - k$ degrees of freedom should be regarded as a fundamental requirement (here $k = 1$ for a linear trend, while $k = 2$ for a quadratic trend). There are two additional parameters (ρ and an intercept parameter) for a total of $k + 2$ parameters. In view of the optimal standard approach, is there any relevant statistical theory suggesting that a t-distribution with $N - k$ degrees of freedom is relevant for assessing the significance of an optimally estimated slope parameter? Without supporting statistical theory and without direct comparison to the optimal estimator, the empirically determined estimator ρ of Equation (3.2) is of questionable value.

Author reply: If the null hypothesis is that no real trend exists, it is fundamentally important that in Monte-Carlo testing of cases of correlated noise with no real trend, the conventionally evaluated "t-statistic" for the regression parameters should follow some known distribution. It would be convenient if this could approximate the t-distribution with $N - k - 1$ degrees of freedom, as it is known to be when ρ is known. Theoretically, the distribution of estimated β parameters changes from a t-distribution with $N - k - 1$ degrees of freedom to something a different when we replace known ρ with an estimate of ρ , and it is not simply a t-distribution with $N - k - 2$ degrees of freedom. Identifying what this distribution should be for the more successful techniques described to estimate ρ may be intractable. Instead, we take the practical approach of assuming a t-distribution with $N - k - 1$ degrees of freedom for the β parameters and concentrate on finding a way of estimating of t that reasonably approximates this.

Author action: No action taken.

Reviewer#2 [8]: In Figure 3, are stationary initial conditions used to initiate the simulation of AR(1) noise? Use of such conditions is important to use for values of ρ close to unity.

Author reply: We agree – this is very important, and we should have mentioned it. Yes, stationary initial conditions were used. The initial random value was matched in mean and standard deviation to the long-term values of the AR(1) noise.

Author action: We have now included in the caption of this figure (now figure 5) a description of the initial conditions used to ensure stationarity.

Reviewer#2 [9]: In the discussion following Equation (4.2), it is not clear to me why inversion of the matrix $X'S-1X$ is regarded as a likely problem. The dimension of this matrix is $(k+1) \times (k+1)$, which is quite small when k is either 1 or 2, as is assumed throughout the manuscript.

Author reply: We agree that it will not be a problem for the simple examples in the original manuscript where k is small, but the manuscript does not assume that the methods are restricted to such small k values; perhaps that was not made clear. Naturally we tested the method with many more variables to ensure stability and general usefulness, and in this testing we occasionally had problems inverting $X'S-1X$. In our ISAP 2016 paper we have 9 explanatory variables, and in some cases tested with this data, $X'S-1X$ was found to be close to singular when estimated ρ was large, while for the same problem inversion of $X'X$ as required for OLS was straightforward.

Author action: We amend "A more likely problem is inversion of $(X'S-1X)$, so the availability of a GLS estimate for a particular type of model is not necessarily guaranteed, particularly if the residual correlation is high" to read "If k is large and correlation is considerable, an occasional problem is inversion of $(X'S-1X)$, so the availability of a GLS estimate for a particular type of model is not necessarily guaranteed." We have revised the final example of 5(c) to discuss the 9 variable ISAP 2016 model, to clarify that the methods are not restricted to just one or two explanatory variables.

Reviewer#2 [10]: There is a substantial literature on using the semivariogram to estimate model parameters, none of which is mentioned in the current manuscript. Two references that come to mind are Interpolation of Spatial Data by M. L. Stein, Springer-Verlag, 1999, and Geostatistics: Modeling Spatial Uncertainty (Second Edition) by J.-P. Chiles and P. Delfiner, Wiley, 2012 (use of these with a citation index should lead to current thinking on semivariogram-based estimators).

Author reply: We agree. Due to the prior experience of the prime author with kriging, many of our initial attempts to solve this problem naturally used a semivariogram approach. While some reasonable results were obtained, our aim was to find a consistent approach that could be applied uniformly to data in one-, two- and three-dimensional space. We found our extension of the Durbin-Watson statistic to random spacing and multi-dimensional space to be preferable if the requirement is a uniform approach for all these cases.

Author action: All testing of spatially distributed data in sections 4 and 5 now includes semivariogram estimation alongside our proposed Durbin-Watson based methods.

Appendix C

Manuscript ID: RSOS-171493

Type: Regular

Title: "How real are observed trends in small correlated datasets?"

Author: Salamon *et al.*

To: Andrew Dunn, Royal Society Open Science Editorial Office

Dear Andrew,

Re: Reply to reviewers on MS # RSOS-181089

Many thanks for having our paper RSOS-181089 reviewed.

We have pleasure in attaching an updated manuscript and our point-by-point response to the comments is given below.

Best regards,

Stephen Salamon et al.

Reviewer#1: The authors have considered my comments satisfactory and changed the title accordingly.

Author reply: We thank the reviewer for the confirmation that the new title is suitable.

Reviewer#3, 1): In this paper a new correlation estimate based on the Durbin-Watson statistic is developed, leading to an improved estimate of autoregression with highly correlated data, thus reducing this risk. These techniques are generalised to randomly located data points in space, through the new concept of the nearest new neighbour path. The authors describe tests on the validity of the GLS schemes, allowing verification of the models employed.

Although the paper could furnish a good contribution to the existing literature, however some significant aspects must be clarified and improved; without these specific requirements the paper cannot be accepted for publication. In particular:

1) It is not specified throughout the paper what are the theoretical hypotheses to apply the new correlation estimate, neither are specified the practical aspects outcoming for applying this new estimate. The above discussion concerns equally spaced data points in one dimensional space.

Author reply: In the case of equally spaced data points in one dimensional space, we use the well-known Durbin-Watson statistic, and in particular refer to its published theoretical study of its statistical distribution. We provide a new estimate of ρ for the first-order autoregressive model and show from published theory that our new estimate will be unbiased for the $\rho=0$ case. Admittedly our tanh adjustment scheme to produce an estimate for other cases is rather more empirical. This was found to be necessary because the literature-based alternative of the beta distribution was found to be numerically impractical for many cases well removed from $\rho=0$. In contrast, our tanh alternative in (3.1) is numerically practical for all cases we have tried, and we show by Monte Carlo testing, e.g. figure 4, that our new estimate has considerably lower bias than other estimates such as MLE, the lag-1 value of the ACF, or the simple Durbin-Watson estimate $\rho = 1 - d/2$. Despite the reduced bias of our new estimate, it has a similar size confidence interval to that of MLE.

Author action: To clarify our equation (3.1) we have amended the end of the last sentence before it, to read "... and expectation $E(d)$ and variance $V(d)$ for the uncorrelated case, from (2.6):"

Reviewer#3, 2): It is not clear how to apply the GLS estimate in a multi-dimensional space. Moreover and most importantly, what is its usefulness in this context? The authors suggest to replace a first order autoregressive model with its equivalent exponential model: however, this is very restrictive. Indeed, geostatistical literature provides several tools and better solutions in this context; for example, anisotropy aspects are not considered by the authors, moreover spatial correlation is well captured by the variogram and several models exist to describe the spatial (or, more generally, spatio-temporal) correlation.

Author reply: We agree that in describing the development our GLS estimate in a multi-dimensional space, we do not have a clear enough statement of the application of the method. We address this by adding a new sub-section 4(e) "Summary of the proposed GLS, for data in multiple dimensions." This describes the proposed procedure. We agree that confining the discussion to an isotropic exponential model is restrictive, but as this method is a generalisation of the Durbin-Watson statistic, it essentially only caters for a single parameter model. Just as the first-order autoregressive model is likely to be the most appropriate single parameter model to estimate from the Durbin-Watson statistic, the well-known, and valid in any number of dimensions, exponential model is likely to be the most appropriate single parameter model to estimate from our nearest new neighbour path definition of sum or squares of residual forward differences. We have not included anisotropy as part of our new model, but now discuss it in the new sub-section 4(e), as well as short- and long-range dependence, and how to identify if alternative models, other than our simple isotropic exponential one, need to be considered.

Author action: We have added a new sub-section 4(e) "Summary of the proposed GLS, for data in multiple dimensions." This describes the proposed method, including testing for validity of the exponential model. The assumptions of our proposed method are discussed, including the possibility of anisotropy.

Reviewer#3, 3): At least in the spatial context, literature must be updated.

Author reply: We agree that the paper did not provide adequate guidance alternatives to the isotropic exponential models discussed, or how to identify if those alternatives needed to be explored.

Author action: The new sub-section 4(e) describing the proposed new procedure includes discussion of testing that may be performed, to identify if alternate models or procedures may be required. This includes references to recent literature on spatial statistics and anisotropy.